# Redirecting electron flow in *Acetobacterium woodii* enables growth on CO and improves growth on formate

Jimyung Moon [1], Anja Poehlein [2], Rolf Daniel [2] & Volker Müller [1]✉

Anaerobic, acetogenic bacteria are well known for their ability to convert various one-carbon compounds, promising feedstocks for a future, sustainable biotechnology, to products such as acetate and biofuels. The model acetogen *Acetobacterium woodii* can grow on $CO_2$, formate or methanol, but not on carbon monoxide, an important industrial waste product. Since hydrogenases are targets of CO inhibition, here, we genetically delete the two [FeFe] hydrogenases HydA2 and HydBA in *A. woodii*. We show that the Δ*hydBA*/*hydA2* mutant indeed grows on CO and produces acetate, but only after a long adaptation period. SNP analyzes of CO-adapted cells reveal a mutation in the HycB2 subunit of the HydA2/HydB2/HydB3/Fdh-containing hydrogen-dependent $CO_2$ reductase (HDCR). We observe an increase in ferredoxin-dependent $CO_2$ reduction and vice versa by the HDCR in the absence of the HydA2 module and speculate that this is caused by the mutation in HycB2. In addition, the CO-adapted Δ*hydBA*/*hydA2* mutant growing on formate has a final biomass twice of that of the wild type.

Carbon monoxide is a colorless, odorless, highly toxic gas for humans as well as most microbes. Some specialized aerobic as well as anaerobic microbes can use carbon monoxide as carbon and energy source for growth[1,2]. Under anaerobic conditions, it is a superb electron donor due to its low redox potential ($E_0'$[CO/$CO_2$] = −520 mV) and some acetogenic as well as methanogenic archaea can couple CO oxidation to $CO_2$ reduction to acetate and methane, respectively[3,4]. Acetogenesis from CO has the biggest change of free energy ($\Delta G_0'$ = −175 kJ/mol) compared to other one carbon substrates such as $CO_2$ and $H_2$ ($\Delta G_0'$ = −95 kJ/mol), formate ($\Delta G_0'$ = −99 kJ/mol) or methanol ($\Delta G_0'$ = −74 kJ/mol). The initial reaction in carbon monoxide metabolism is its oxidation according to:

$$CO + H_2O \rightarrow CO_2 + 2H^+ + 2e^- \qquad (1)$$

The low redox potential of the CO/$CO_2$ couple allows for the reduction of the electron acceptor ferredoxin[5] and in acetogenic bacteria, redox potentials ($E_0'$) of −380 to −559 mV have been determined for different ferredoxins[6,7]. Reduced ferredoxin fuels the respiratory enzymes of acetogens, the energy-converting hydrogenase (Ech) or the ferredoxin:NAD-oxidoreductase (Rnf) that couple electron transfer from reduced ferredoxin to protons or NAD, respectively, to vectorial ion ($H^+$, $Na^+$) transport across the cytoplasmic membrane and the so established electrochemical ion gradient drives the synthesis of ATP[8,9]. Thus, acetogenesis from CO has the highest ATP yield of all acetogenic one-carbon substrates. In the model acetogenic, Rnf-containing bacterium *Acetobacterium woodii*, it is calculated to 1.5 ATP/acetate compared to only 0.3 mol ATP/acetate when grown on $H_2$ and $CO_2$.

Acetogenic bacteria have gained much interest in recent years since they are considered as prime production platforms in a novel, sustainable biotechnology that no longer uses sugar-derived feedstocks but one-carbon compounds such as CO, $CO_2$, formate or methanol[4,10–13]. The acetate produced can then be the starting point for all synthetic reactions catalyzed by various microbes to produce all the compounds that are produced today from glucose by biological

[1]Department of Molecular Microbiology & Bioenergetics, Institute of Molecular Biosciences, Johann Wolfgang Goethe University, Max-von-Laue Str. 9, Frankfurt, Germany. [2]Göttingen Genomics Laboratory, Institute for Microbiology and Genetics, Georg August University, Grisebachstr. 8, Göttingen, Germany. ✉e-mail: vmueller@bio.uni-frankfurt.de

processes or from petrochemicals chemically[14]. Alternatively, metabolic engineering efforts to broaden the portfolio of products formed by acetogenic bacteria have increased dramatically[15,16]. However, since the ATP yield of acetogenesis is rather low, the products that can be formed from $H_2$ and $CO_2$ by acetogens naturally or by metabolic engineering are rather restricted[17]. Therefore, the only industrial application that uses acetogenic bacteria to date is the production of ethanol, isopropanol and acetone from syngas, a gas mixture that contains CO as well as $CO_2$ and $H_2$[18,19].

Although many if not all acetogenic bacteria are able to produce acetate from $CO_2$[2,20–22], not every species can grow on CO. This is also true for the model acetogen *A. woodii*[23]. Since hydrogenases are generally very sensitive to CO[24], we hypothesized that CO inhibition of hydrogenases causes the inability to grow on CO. This hypothesis is somewhat corroborated by the observation that growth of *A. woodii* is possible on CO and formate, a pathway that does not involve hydrogenases[23]. *A. woodii* has two hydrogenases[25]. One (HydA2) is a subunit of the hydrogen-dependent $CO_2$ reductase (HDCR) that reduces $CO_2$ to formate with electrons derived directly from hydrogen oxidation[26]. This remarkable enzyme has a long chain of small, iron-sulfur-containing proteins (HycB2 and HycB3) that make a long electron wire. A FeFe-hydrogenase module (HydA2) and a formate dehydrogenase module (FDH) sit on the wire-like light bulbs in a fairy light[27]. Interestingly, the purified enzyme can also use reduced ferredoxin as reductant[26]. The second enzyme is the electron-bifurcating hydrogenase HydABC that reversibly oxidizes hydrogen with the simultaneous reduction of NAD and ferredoxin[28]. In the absence of molecular hydrogen, the enzyme produces $H_2$ required for the reduction of $CO_2$ to formate by the HDCR from the soluble electron carriers NADH and reduced ferredoxin[28,29]. During growth on formate, oxidation of formate by HDCR yields hydrogen gas and as above, $H_2$ is oxidized by the HydABC hydrogenase to provide NADH and reduced ferredoxin required for $CO_2$ and formate reduction[30]. Acetogenesis from formate gives the same amount of ATP as acetogenesis from $H_2$ and $CO_2$ but is challenged by the escape of hydrogen into the environment. A simplified model of acetogenesis from CO or formate that highlights the role of the two hydrogenases in *A. woodii* and the respiratory chain is shown in Fig. 1.

Here, we show that the *A. woodii* mutant lacking both hydrogenases can grow on carbon monoxide, and these cells also grow to much higher yields than the wild type strain with formate as energy and carbon source. The results will facilitate the using of *A. woodii* as a platform for the production of added-value compounds from carbon monoxide.

## Results

### Generation of the hydrogenase-free mutant Δ*hydBA*/*hydA2*

The genome of *A. woodii* encodes two hydrogenases, the electron-bifurcating hydrogenase HydABC and the HydA2 hydrogenase subunit of the HDCR[25]. To generate a double mutant (Δ*hydBA*/*hydA2*), we took advantage of the previously described Δ*hydBA* mutant[31] that we used as a background to delete *hydA2* by allelic exchange mutagenesis (Supplementary Fig. 1). First, we analyzed the growth phenotype of the Δ*hydBA*/*hydA2* mutant on fructose and one-carbon compounds. Unlike the Δ*hydBA*/*hdcr* mutant[32], the Δ*hydBA*/*hydA2* mutant could grow on fructose alone with a growth rate and final $OD_{600}$ similar to the Δ*pyrE* strain (Supplementary Fig. 2a). The mutant converted $22.9 \pm 1.5$ mM fructose to $58.6 \pm 3.1$ mM acetate with a fructose:acetate ratio of 1:2.6. As minor product, $1.5 \pm 0.2$ mM ethanol was produced. These data demonstrate that hydrogen cycling is not essential to couple the oxidative and the reductive branches of acetogenesis from fructose and that an alternative reductant was used for $CO_2$ reduction. As expected, the hydrogenase-free mutant did not grow on $H_2$ and $CO_2$ (Supplementary Fig. 2b).

### Adaptation of the Δ*hydBA*/*hydA2* mutant to grow on CO

First, we tried to adapt the Δ*hydBA*/*hydA2* mutant to grow on CO. There was no immediate growth when transferred to medium that contained 25% CO as sole carbon and energy source, but after 6 months we observed growth to a final $OD_{600}$ of 0.2. After transferring the culture three times (stationary phase cultures, 10% inoculum), the mutant could grow on 25% CO to a final $OD_{600}$ of 0.4 with a growth rate of $0.02\,h^{-1}$ (Fig. 2a). After the fifth transfer, growth of the mutant on 25% CO was enhanced with a final $OD_{600}$ of 1.0 and a growth rate of $0.05\,h^{-1}$; the growth rate and yield did not increase further upon prolonged incubation with CO. There was no growth in the absence of CO. Then, we tested the CO tolerance of the mutant. The mutant also tolerated 50, 75, and 100% CO in the headspace and grew to similar final $OD_{600}$ and with similar growth rates as with 25% CO (Fig. 2b), indicating that the CO concentration was not growth limiting. After growth, similar amounts of acetate (21 to 25 mM) were produced, regardless of the CO concentrations. Other products, such as ethanol, lactate, and formate, were not produced. The CO-adapted Δ*hydBA*/*hydA2* mutant pre-grown on fructose could grow on CO without a lag phase, indicating the occurrence of mutations to cause growth on carbon monoxide.

### Mutations in the genome of Δ*hydBA*/*hydA2* mutant during adaptation on CO

To identify mutations that occurred during adaptation on CO, single nucleotide polymorphism (SNP) analyses were performed. The DNA sequencing revealed that 100% of the population of the mutant had a mutation in the *hycB2* gene after adaptation on CO (Table 1). Furthermore, 76.4% of the population after the first adaptation and 81.2% of that after tenth transfer showed mutation in the *modC2* gene which affects transport of the key cofactor of the FDH, molybdenum. Mutation was also seen in the *fdhC* gene encoding a formate transporter: this was observed in 66.8% of the population after the first adaptation as well as 78.9% of the population after the tenth transfer. A small portion of the population after tenth transfer had mutations in the genes encoding the CODH/ACS complex as well.

### Conversion of CO in resting cells of the Δ*hydBA*/*hydA2* mutant

Next, we analyzed products formed from CO in resting cells of the double mutant. For these experiments, cells were grown with 50% CO until stationary growth phase and after harvesting, cell suspensions were prepared as described in Methods. When 50% CO was provided as substrate, resting cells of the Δ*hydBA*/*hydA2* mutant produced $18.3 \pm 1.0$ mM acetate and $17.9 \pm 0.0$ mM formate from $95.2 \pm 4.7$ mM CO in the presence of 60 mM $KHCO_3$ with a CO:acetate:formate ratio of 5.3:1:1 (Fig. 3a), whereas growing cells only produced acetate from CO. In previous study, we found that bicarbonate uncoupled ATP formation in resting cells of the thermophilic acetogen *Thermoanaerobacter kivui*[33]. Since ATP is required for the synthesis of formyl-tetrahydrofolate from formate and tetrahydrofolate, acetogenesis from formate was abolished and formate was only converted to $H_2$ and $CO_2$ by the HDCR[34]. Consistently, when 300 mM $KHCO_3$ was present in the cell suspensions of *A. woodii*, acetogenesis from CO was abolished and only formate ($25.6 \pm 1.0$ mM) was produced from $28.9 \pm 2.3$ mM CO with a CO:formate ratio of 1.1:1. (Fig. 3b). In agreement with cell suspension experiments, the Δ*hydBA*/*hydA2* mutant did not grow on CO in the presence of 300 mM $KHCO_3$.

In the absence of bicarbonate, acetate should be the only product from CO, but formate ($10.9 \pm 0.7$ mM) was produced alongside acetate ($9.4 \pm 0.6$ mM) from CO ($51.6 \pm 2.9$ mM) with a CO:acetate:formate ratio of 5.5:1:1.1 (Fig. 3c). Acetogenesis from $H_2$ and $CO_2$ or formate is $Na^+$-dependent, due to the action of the $Na^+$-translocating Rnf complex[30,35] (cf. Figure 1), and in the absence of NaCl, formate was only converted to $H_2$ and $CO_2$ and vice versa[26,36]. In the absence of NaCl, resting cells of the Δ*hydBA*/*hydA2* mutant converted CO

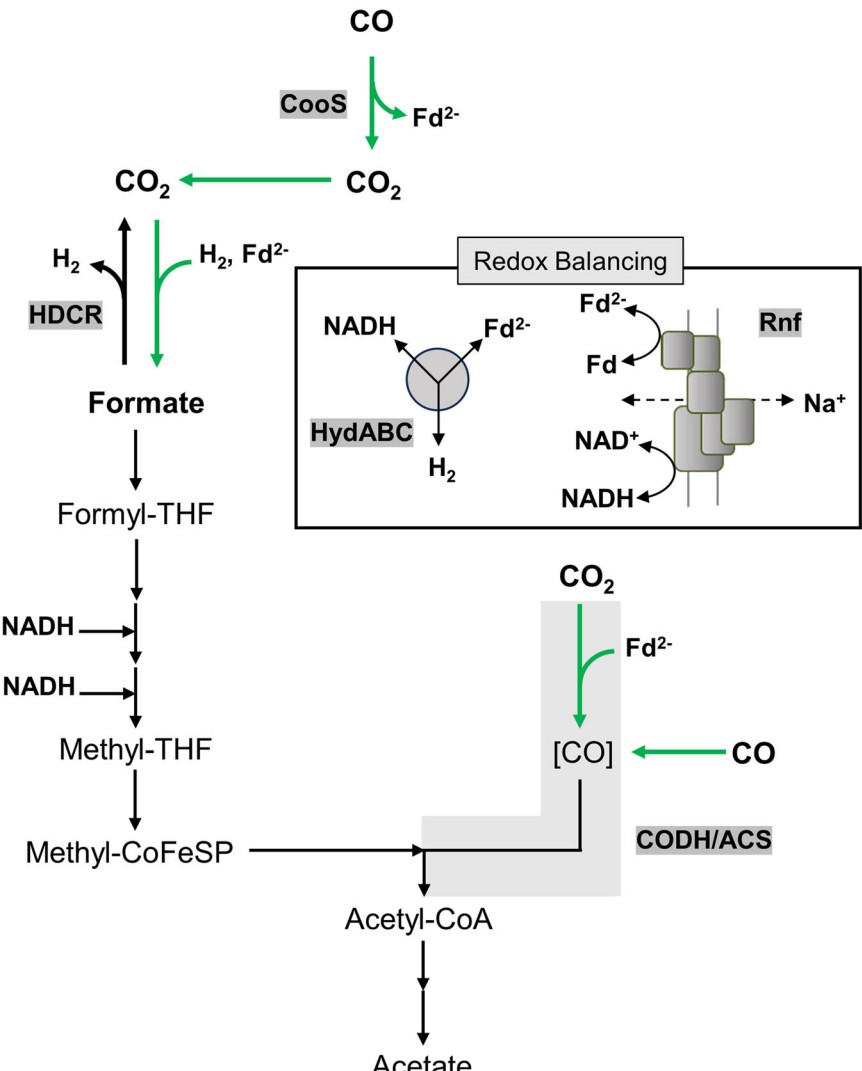

**Fig. 1 | Acetogenesis from formate and CO in *A. woodii*.** During growth on formate, three-quarters of formate are oxidized to $CO_2$ and $H_2$ by the HDCR (black arrow). $H_2$ is oxidized by the HydABC hydrogenase to reduce Fd and $NAD^+$, 0.5 mol of $Fd^{2-}$ is oxidized by the Rnf complex to reduce $NAD^+$. Carbon dioxide is reduced to enzyme-bound carbon monoxide ([CO]) by the CO dehydrogenase of the CODH/ACS complex. During growth on carbon monoxide (CO) enters the CODH/ACS and is condensed with the methyl group and CoA to acetyl-CoA (green arrows). Electrons for the reduction of $CO_2$ to the methyl group are generated by the monofunctional CO dehydrogenase CooS (green arrows). CO oxidation yields $Fd^{2-}$ only and $Fd^{2-}$ is a major electron donor for anabolic and catabolic reactions. $H_2$ and NADH are provided by the HydABC hydrogenase and Rnf complex. Source data are provided as a Source Data file.

(35.3 ± 4.2 mM) to formate only (31.9 ± 0.8 mM) with a CO: formate ratio of 1.1:1 (Fig. 3d), demonstrating that acetogenesis from CO also requires $Na^+$/the Rnf complex.

### Fd-dependent FDH activity in the CO-adapted Δ*hydBA/hydA2* mutant

The above-mentioned experiments can be interpreted to mean that the HDCR is still active in the Δ*hydBA/hydA2* strain by using ferredoxin as electron carrier, as recently demonstrated for a Δ*hydA2* mutant of *T. kivui*[37]. To verify that the Δ*hydBA/hydA2* mutant has a functional Fd-dependent HDCR we performed enzymatic assays in cell-free extracts. For these experiments, the Δ*pyrE* strain, Δ*hydBA/hdcr* and CO-adapted Δ*hydBA/hydA2* mutants were grown on 20 mM fructose and after harvesting, cell-free extracts were prepared as described in Methods. The Δ*pyrE* and Δ*hydBA/hydA2* strains catalyzed formate-dependent reduction of methyl viologen, indicating the presence of a functional formate dehydrogenase module; this activity was not observed in the Δ*hydBA/hdcr* strain, as expected. Cell-free extract of the Δ*hydBA/hdcr* and Δ*hydBA/hydA2* mutants did neither catalyze formate-dependent hydrogen evolution nor hydrogen-dependent formate production from $CO_2$. In contrast, cell-free extract of the Δ*pyrE* strain evolved hydrogen from formate with a rate of 148.7 ± 17.8 mU/mg and produced formate from $H_2$ and $CO_2$ with a rate of 143.2 ± 10.3 mU/mg. The cell-free extract of the Δ*pyrE* strain was also able to perform formate-dependent reduction of Fd with an activity of 4.1 ± 0.8 mU/mg. Surprisingly, the cell-free extract of the Δ*hydBA/hydA2* mutant showed 6.7 times higher activity (27.3 ± 4.6 mU/mg) compared to those of the Δ*pyrE* strain. $Fd^{2-}$-dependent production of formate was even 25-times higher (55.3 ± 13.4 mU/mg) compared to the Δ*pyrE* strain (2.2 ± 0.4 mU/mg). The cell-free extracts of the Δ*hydBA/hdcr* mutant did not show any Fd-dependent FDH activities.

### Growth of the CO-adapted Δ*hydBA/hydA2* mutant on formate

The above-mentioned experiments are in line with the hypothesis that the deletion of the hydrogenase module renders the HDCR active by using reduced ferredoxin instead of molecular hydrogen as electron donor, as observed before with *T. kivui*[37]. If this is true, the double mutant should also grow on formate. Moreover, since all the reducing

equivalents from formate oxidation are channeled into the CO$_2$-reduction pathway directly by soluble ferredoxin without a loss of electrons into the environment by evolution of molecular hydrogen and since the only enzyme that catalyzes oxidation of reduced

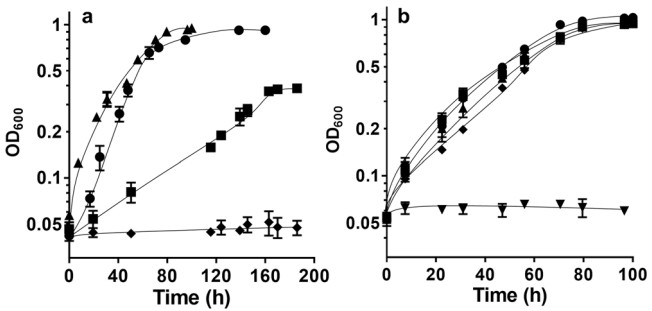

**Fig. 2 | Adaptation of the Δ*hydBA/hydA2* mutant to growth on CO.** Cells of the Δ*hydBA/hydA2* mutant grown on 20 mM fructose were transferred to 5 mL bicarbonate-buffered complex medium in 16 mL Hungate tubes under a N$_2$/CO$_2$/CO atmosphere (2 bar, 56:19:25, v/v/v). **a** Optical densities of the Δ*hydBA/hydA2* mutant after the third (squares), fifth (circles) and tenth transfer (triangles) on 25% CO are presented. The control strain Δ*pyrE* (diamonds) did not grow on 25% CO. **b** After the tenth transfer, the Δ*hydBA/hydA2* mutant was grown on 0 (inverted triangles), 25 (squares), 50 (circles), 75 (triangles) and 100% CO (diamonds). Each data point presents a mean ± standard deviation (SD); *n* = 3 independent experiments. Source data are provided as a Source Data file.

ferredoxin is the energy-conserving Rnf complex[35], growth yields on formate should be increased in the double mutant. The CO-adapted Δ*hydBA/hydA2* mutant did not grow on formate in bicarbonate-buffered complex medium, but in bicarbonate-free, phosphate-buffered medium it grew to an OD$_{600}$ of 0.2. After three transfers, the final OD$_{600}$ increased to 0.4. After ten transfers, the mutant grew on 100 mM formate with a similar growth rate as the Δ*pyrE* strain but produced twice the amount of biomass (final OD$_{600}$ of 0.9) (Fig. 4a). Noteworthy, formate conversion to acetate in the mutant was slower and formate was not completely consumed, whereas the Δ*pyrE* strain consumed formate completely in 60 h (Figs. 4b, c). No other reduced product was observed and the formate:acetate ratio in both strains was 1:3.8, implying that the mutant performs homoacetogenesis from formate. The difference of the final OD$_{600}$ of the Δ*pyrE* and Δ*hydBA/hydA2* mutant was even bigger at low formate concentrations, arguing for an advantage of Fd-dependent HDCR for higher net ATP gain.

Growth of the mutant on formate was observed only in bicarbonate-free phosphate-buffered medium and, therefore, we analyzed the effect of bicarbonate and phosphate on growth. Growth on formate in phosphate-buffered medium was not affected by adding 20 mM bicarbonate, but growth was drastically impaired by higher bicarbonate concentrations. At 40 mM bicarbonate, the final OD$_{600}$ was decreased by 60%, at 100 mM by 90% and at 200 mM, it was completely inhibited (Supplementary Fig. 3). In contrast, adding phosphate to bicarbonate-buffered medium did not enhance the growth of the mutant on formate, indicating that bicarbonate inhibits growth on formate.

## Table 1 | Single nucleotide polymorphisms of the Δ*hydBA/hydA2* mutant after adaptation to CO

| Position (bp) | Mutation | Δ*hydBA/hydA2* | | | Amino acid change | Gene plus → or minus ← strand | Annotation |
|---|---|---|---|---|---|---|---|
| | | Before adaptation | After adaptation (6 month after on 25% CO, OD 0.2) | 10th transfer | | | |
| **925,621** | **A → T** | | 66.8% | 78.9% | **K21I (AAA → ATA)** | ***fdhC* →** | **formate/nitrite transporter FdhC** |
| **950,586** | **G → C** | 36.5% | 100% | 100% | **R165P (CGG → CCG)** | ***hycB2* →** | **hydrogenase Fe-S subunit HycB2** |
| 1,004,414 | G → A | | 44.9% | 33.1% | P84S (CCG → TCG) | *lctA* ← | regulator for LDH/ETF complex |
| 1,004,442 | Δ15 bp | | | 6.2% | coding (208-222/ 711 nt) | *lctA* ← | regulator for LDH/ETF complex |
| 1,004,462 | C → T | | 12.8% | | G68S (GGT → AGT) | *lctA* ← | regulator for LDH/ETF complex |
| **1,243,519** | **C → T** | | | 5.3% | **P236L (CCT → CTT)** | ***acsA* →** | **CO dehydrogenase, catalytic subunit AcsA** |
| **1,245,411** | **T → A** | | 10.1% | | **I225N (ATC → AAC)** | ***cooC2* →** | **CO dehydrogenase nickel-insertion accessory protein CooC2** |
| **1,245,718** | **C → T** | | | 8.7% | **P47L (CCT → CTT)** | ***acsB1* →** | **acetyl CoA synthase catalytic subunit AcsB** |
| **1,245,999** | **G → A** | | | 21.8% | **V141I (GTA → ATA)** | ***acsB1* →** | **acetyl CoA synthase catalytic subunit AcsB** |
| 2,849,933 | A → T | | | 34.9% | intergenic (-79/ + 265) | *nifJ* ← / ← *Awo_c24340* | pyruvate:ferredoxin oxidoreductase NifJ/ hypothetical protein |
| 2,884,514 | G → A | | 44.6% | 39.3% | L264F (CTT → TTT) | *fliG* ← | flagellar motor switch protein FliG |
| 2,884,844 | G → A | | 30.1% | 42.3% | R154W (CGG → TGG) | *fliG* ← | flagellar motor switch protein FliG |
| 2,939,053 | (ATGGCTCTTT)$_{1→2}$ | | | 8.3% | coding (959/1014 nt) | *fliM* ← | flagellar motor switch protein FliM |
| 2,940,413 | G → A | | 13.1% | | P126L (CCA → CTA) | *motB* ← | flagellar motor protein MotB |
| **3,227,278** | **A → C** | | 76.4% | 81.2% | **I29S (ATT → AGT)** | ***modC2* ←** | **molybdate/tungsten ABC transport system ATP binding protein ModC2** |

Bold: Genes involved in the Wood-Ljungdahl pathway.

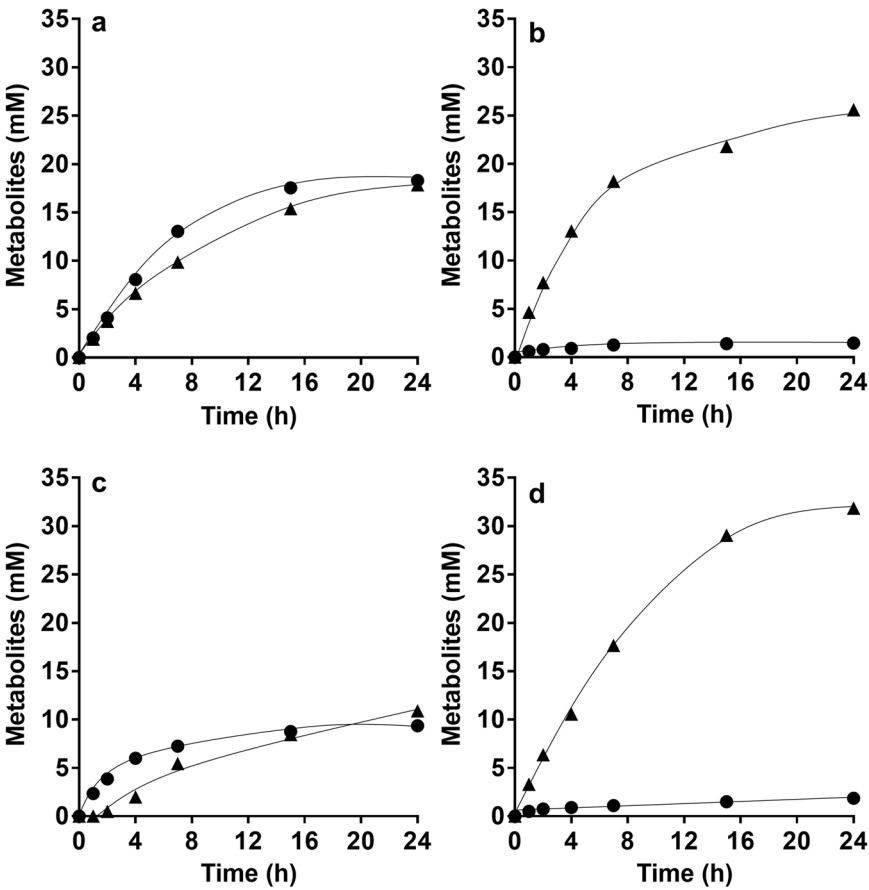

**Fig. 3 | Conversion of CO in resting cells of the ΔhydBA/hydA2 mutant.** Cells of the CO-adapted ΔhydBA/hydA2 mutant were grown in bicarbonate-buffered complex media under a $N_2/CO_2/CO$ atmosphere (2 bar, 40:10:50, v/v/v) and harvested in the early stationary growth phase. Cell suspensions were prepared in 10 mL of cell suspension buffer (50 mM imidazole, 20 mM $MgSO_4$, 20 mM KCl, 20 mM NaCl, pH 7.0) in 120 mL serum flasks to a final protein concentration of 1 mg/mL with 60 mM $KHCO_3$ under 2 bar of a $N_2/CO_2/CO$ (40:10:50, v/v/v) atmosphere (**a**), 300 mM $KHCO_3$ under 2 bar of a $N_2/CO_2/CO$ (40:10:50, v/v/v) atmosphere (**b**), under $CO_2$/bicarbonate-depleted conditions with 2 bar of a $N_2/CO$ (50:50, v/v) atmosphere (**c**), or under $Na^+$-depleted conditions with 60 mM $KHCO_3$ under 2 bar of a $N_2/CO_2/CO$ (40:10:50, v/v/v) atmosphere (**d**). The contaminating $Na^+$ concentration was 0.1 mM. Acetate (circles) and formate (triangles) were determined. Each data point presents a mean; $n$ = 2 independent experiments. Source data are provided as a Source Data file.

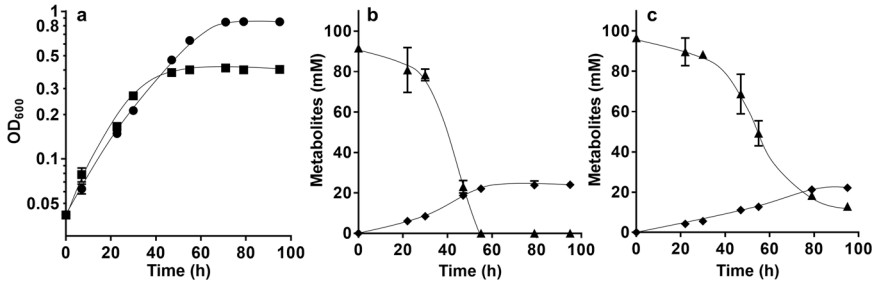

**Fig. 4 | The CO-adapted ΔhydBA/hydA2 mutant grows on formate. a** Cells of the ΔpyrE (squares) and the CO-adapted ΔhydBA/hydA2 mutant (circles) were grown on 100 mM formate in phosphate-buffered complex medium under a 100% $N_2$ atmosphere. Formate (triangles) and acetate (diamonds) were determined in the ΔpyrE (**b**) and ΔhydBA/hydA2 mutant (**c**) cultures. Each data point presents a mean ± SD; $n$ = 3 independent experiments. Source data are provided as a Source Data file.

## Conversion of formate in resting cells of the CO-adapted ΔhydBA/hydA2 mutant

We further analyzed formate conversion and ATP formation in resting cells of the CO-adapted ΔhydBA/hydA2 mutant. For these experiments, cells were grown on 100 mM formate in phosphate-buffered complex medium until stationary growth phase, harvested by centrifugation and cell suspensions were prepared as described in Methods. Here, we added higher concentrations of formate (250 mM) for a clearer comparison since formate conversion in resting cells was reported to be extremely fast[36]. When 250 mM formate was added, resting cells of the ΔpyrE strain converted 247.4 ± 1.1 mM formate to 58.8 ± 0.4 mM acetate with a formate consumption rate of 792.9 ± 27.7 nmol $min^{-1}$ $mg^{-1}$ and acetate formation rate of 158.8 ± 19.4 nmol $min^{-1}$ $mg^{-1}$ (Fig. 5a). While formate was completely consumed in resting cells of the ΔpyrE strain in 4 hours, formate was not completely consumed in resting cells of the ΔhydBA/hydA2 mutant even after 24 hours (Fig. 5b). Only 129.2 ± 29.6 mM formate were converted to 31.4 ± 4.9 mM acetate and the formate consumption rate (222.5 ± 20.0 nmol $min^{-1}$ $mg^{-1}$) and

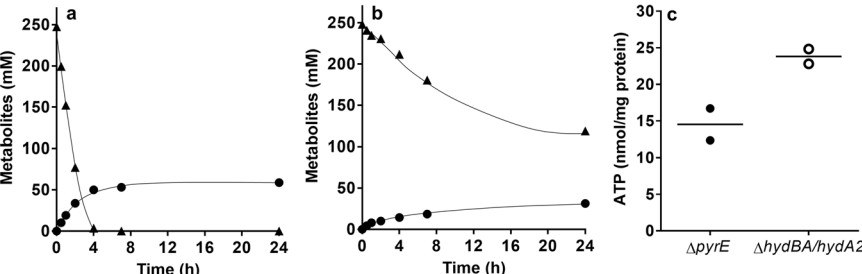

**Fig. 5 | Conversion of formate in resting cells of the CO-adapted Δ*hydBA/hydA2* mutant.** Cells of the Δ*pyrE* (**a**) and the CO-adapted Δ*hydBA/hydA2* mutant (**b**) were grown in phosphate-buffered complex media under a 100% N$_2$ atmosphere with 100 mM formate until the early stationary growth phase. After harvesting, the cell suspensions were prepared in 10 mL of cell suspension buffer (50 mM imidazole, 20 mM MgSO$_4$, 20 mM KCl, 20 mM NaCl, 20 mM KHCO$_3$, pH 7.0) in 120 mL serum flasks to a final protein concentration of 1 mg/mL. 250 mM formate was added to the cell suspensions as carbon and energy source. Formate (triangles) and acetate (circles) were determined at each time point. Each data point indicates a mean; $n = 2$ independent experiments. **c** After 30 min of incubation, ATP contents in resting cells of Δ*pyrE* (filled circles) and the CO-adapted Δ*hydBA/hydA2* mutant (empty circles) were determined. Each line indicates a mean; $n = 2$ independent experiments. Source data are provided as a Source Data file.

acetate formation rate ($80.4 \pm 9.1$ nmol min$^{-1}$ mg$^{-1}$) were only 28% and 51% of the Δ*pyrE* strain, respectively. However, the ATP content in resting cells of the Δ*hydBA/hydA2* mutant after 30 min was 70% higher ($23.8 \pm 1.4$ nmol/mg protein) compared to the Δ*pyrE* strain ($14.5 \pm 3.1$ nmol/mg protein) (Fig. 5c).

## Discussion

Carbon monoxide is not only toxic to many aerobic organisms but also many anaerobes. The acetogen *A. woodii* cannot grow on CO which is in contrast to, for example, *T. kivui*, the phylogenetically related *Eubacterium limosum* or *Clostridium autoethanogenum*. *T. kivui* has the electron-bifurcating [FeFe] hydrogenase HydABC that is apparently not essential for growth on CO and no Rnf complex but a [NiFe] hydrogenase, the Ech complex. *E. limosum* is very similar to *A. woodii*, has only two hydrogenases, both of the [FeFe] type. One is the electron-bifurcating hydrogenase HydABC[38], the other most likely part of a hydrogenase-formate dehydrogenase complex[39,40]. In contrast to *A. woodii*, *E. limosum* does not encode a monofunctional CO dehydrogenase but only the CODH/ACS complex[38] but the *E. limosum* KIST strain was isolated with CO as carbon and electron source. It is not excluded that mutations occurred in the hydrogenase-formate dehydrogenase complex during the first isolation to allow efficient utilization of ferredoxin; formate production from H$_2$ + CO$_2$ by the complex was stimulated by ferredoxin, indicating that the enzyme can also use ferredoxin as electron carrier[37]. *C. autoethanogenum* contains [FeFe] as well as [NiFe] hydrogenases and an Rnf complex[41]. Thus, genome comparisons of CO-insensitive and sensitive strains do not necessarily reveal the molecular basis for CO insensitivity.

*A. woodii* did grow on CO when it was provided together with formate. Under these conditions, formate served as electron acceptor for electrons derived from CO oxidation. This was taken as indication that the HDCR is the/one target of CO inhibition[23]. Indeed, biochemical analysis revealed CO inhibition of the purified HDCR but, astonishingly, CO inhibition was fully reversible[42]. When formate was given together with CO, inhibition of growth could be overcome[23]. Under these conditions, formate is only reduced with NADH as reductant. NAD$^+$ is reduced with reduced ferredoxin as reductant by the Rnf complex with concomitant synthesis of ATP by the respiratory chain; the HydABC hydrogenase is not involved in electron flow. Carbon monoxide metabolism in the Δ*hydBA/hydA2* mutant is postulated to involve the HDCR[43]. In this case, reduced ferredoxin is assumed to donate electrons to the HDCR via the nanowire proteins HycB2 and/or HycB3, as previously assumed for the HDCR from *T. kivui*[37]. Indeed, the HDCR purified from *A. woodii* can use reduced ferredoxin for carbon dioxide reduction to formate, albeit with only 6% of the rate compared to H$_2$ in the wild type enzyme[26]. The formate produced is then reduced

with two mol NADH to a methyl group that is then condensed with another CO to acetyl-CoA. The ATP yield is a little lower (80%, 1.2 ATP/mol of acetate) compared to the wild type metabolism (Figs. 6a, b).

Although cells of the hydrogenase-free mutant Δ*hydBA/hydA2* grew on CO, they did only after a lag phase of six months. This long lag phase may not be specific to the substrate CO but such a long lag phase is also observed when shifting cells of *Methanosarcina barkeri* from an energy rich substrate like trimethylamine to a poor substrate like acetate and may involve adjustment of regulatory circuits[44]. After adaptation of *Thermococcus onnurineus* to CO by ALE, one mutation was found in a potential regulatory protein. In addition, deletion of hydrogenase genes, formate dehydrogenase and formate transporter was observed[45]. In *A. woodii*, it may also involve genetic adaptation to circumvent CO toxicity to other growth essential proteins outside the CO$_2$ reduction pathway and the Rnf complex or hydrogenase, but we did not find such mutations. ALE of *E. limosum* to growth at high CO concentrations revealed mutations in *acsA*, the CO dehydrogenase subunit of the CODH/ACS as well as in *cooC*[46]. We also found mutations in *acsA* and *cooC* as well as in *acsB*, but only in small fractions of the population. Our SNP analysis revealed that 100% of the population had a mutation in HycB2, the ferredoxin-like subunit of the HDCR that, together with HycB3, makes the long nanowire to which the hydrogenase and formate dehydrogenase modules are hooked up. In addition, a mutation in the formate transporter FdhC as well as in a potential molybdate/tungsten transporter (required for formate dehydrogenase) occurred. The latter two may contribute to higher growth yields of the mutant on formate. However, the mutation in HycB2 is the most prominent. Very recently, we identified mutations in *acsA*, *cooC*, *hydA1* (electron bifurcating hydrogenase) and two genes encoding subunits of the multisubunit Ech hydrogenase in CO-adapted strains of *T. kivui*. Most striking was a mutation in *hycB3* in 100% of the population[47]; HycB3 is the second subunit that together with HycB2 makes the polyferredoxin-like nanowire of the HDCR, and the reported mutation resulted in a loss of the oligomeric structure. Instead, only a single, heterohexameric complex was found[27]. The breaking of the filament into single, functional modules may improve the interaction with ferredoxin. It is tempting to speculate that the mutation in *hycB2* has the same effect as the mutation in *hycB3*.

When the Δ*hydBA/hydA2* mutant was transferred to media containing only formate as energy source, growth did not occur. Also, we were not able to adapt the Δ*hydBA/hydA2* double mutant to growth on formate (tested for 6 months). However, the CO-adapted strain Δ*hydBA/hydA2* grew on formate immediately, indicating again that not CO toxicity but HDCR adaptation to ferredoxin is the cause of the long lag phase. Although growth on formate was poor at the beginning, further rounds of transfer increased the growth rate to a value

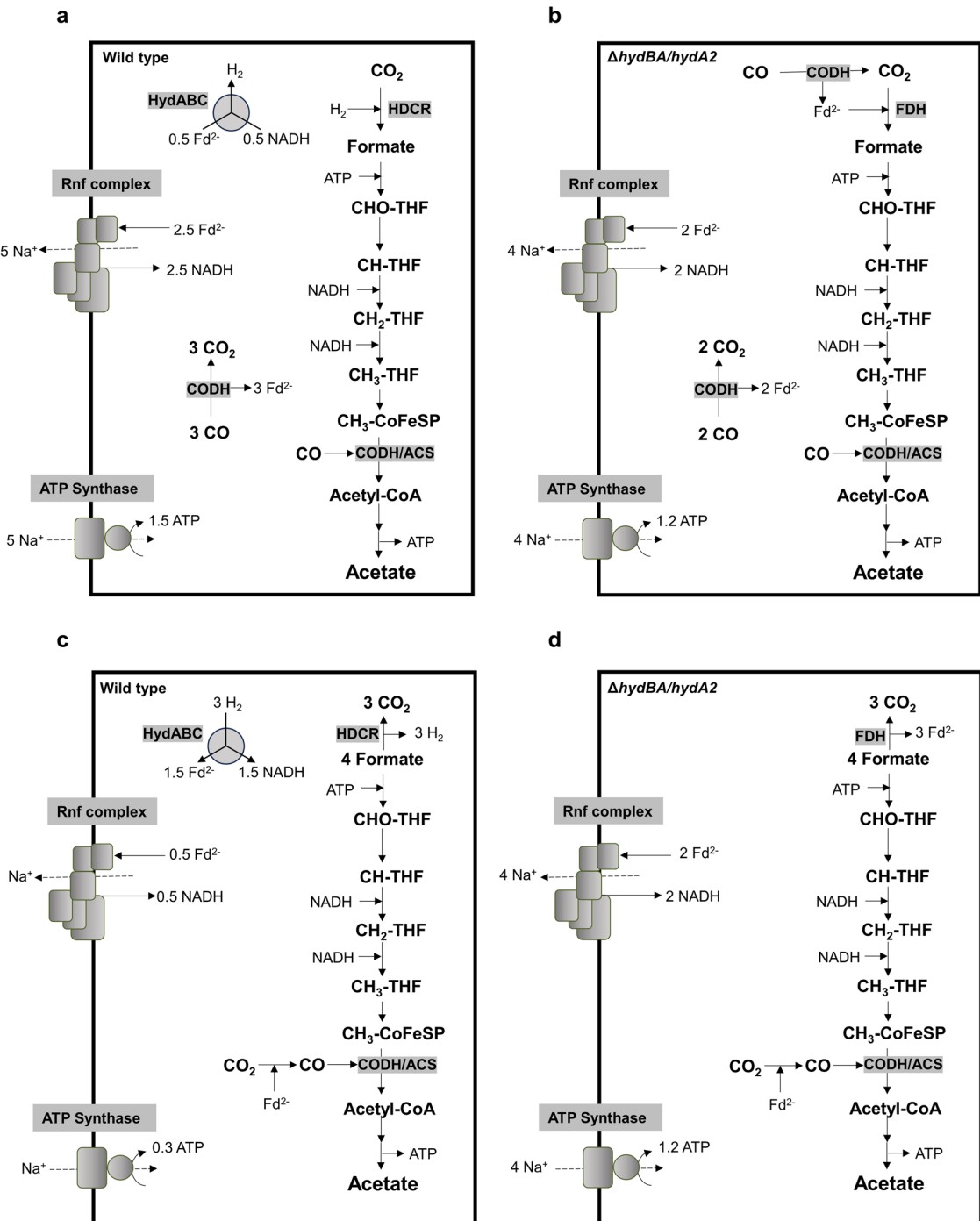

**Fig. 6 | Biochemistry and bioenergetics of acetogenesis from CO (a, b) or formate (c, d) of *A. woodii*.** Panel (**a**) and (**c**) depict models of the wild type, whereas panel (**b**) and (**d**) present models of the Δ*hydBA/hydA2* mutant. Fd, ferredoxin; THF, tetrahydrofolate; CODH, CO dehydrogenase; ACS, acetyl-coenzyme A synthase; CoFeSP, corrinoid iron-sulfur protein. The stoichiometry of the ATP synthase is 3.3 Na⁺/ATP[62] and for the Rnf complex a stoichiometry of 2 Na⁺/2 e⁻ is assumed.

comparable to the wild type. Strikingly, the Δ*hydBA/hydA2* mutant produced twice as much biomass as the wild type. This is expected since formate oxidation in the wild type leads to the production of gaseous H₂ that has to be recaptured and then oxidized by the electron bifurcating hydrogenase HydABC. Only 0.5 mol of reduced ferredoxin is oxidized by the Rnf complex coupled to ATP synthesis of only 0.3 mol of ATP/mol of acetate (Fig. 6c). In stark contrast, in the Δ*hydBA/hydA2* mutant formate is oxidized with simultaneous reduction of ferredoxin (see Fig. 6d). All the reduced ferredoxin is then oxidized via the energy-conserving Rnf complex, this increases the amount of Na⁺ translocated and the amount of ATP produced (1.2 mol/mol of acetate) by 300%. This not only increases the amount of biomass produced by 100% but also makes formate a superior soluble CO₂ equivalent. Acetogenesis from H₂ and CO₂ only yields 0.3 mol ATP/mol of acetate. Chemical and biological methods to produce formate from H₂ + CO₂ are available[26,48–51] and may then be used by the Δ*hydBA/hydA2* mutant to increase yields of added-value compounds such as acetone dramatically. The higher yield during growth on formate is a big step in overcoming energetic barriers in acetogenic one-carbon conversion. It should also be mentioned in this connection that this strain is superior in mixotrophic fermentation, such as syngas fermentation or fermentation of formate and CO.

## Methods

### Strains and cultivation

*A. woodii* wild type (DSM1030) was obtained from the Deutsche Sammlung von Mikroorganismen und Zellkulturen (DSMZ; Braunschweig, Germany). The Δ*pyrE* strain and the Δ*hydBA/hdcr* mutant were described before[31,32,35]. The hydrogenase-free double mutant Δ*hydBA/hydA2* was generated in this study. All strains were routinely cultivated under anoxic conditions at 30 °C in bicarbonate-buffered (60 mM) complex medium under a $N_2/CO_2$ atmosphere (80:20, v/v)[52]. For growth of the CO-adapted Δ*hydBA/hydA2* mutant on formate, phosphate-buffered (60 mM) complex medium[23] was used, and the gas phase was changed to 100% $N_2$. As growth substrates, 20 mM fructose, 20–500 mM sodium formate or 25–100% CO (2 bar; 100 mL of headspace) was added. Growth was monitored by following the optical density at 600 nm ($OD_{600}$).

### Generation of *A. woodii* Δ*hydBA/hydA2* mutant

To generate the Δ*hydBA/hydA2* mutant, the plasmid pMTL84151_JM_dhydA2 was constructed in *E. coli* HB101 (Promega, Madison, WI, USA) and transformed into the *A. woodii* Δ*hydBA* strain[31]. The plasmid pMTL84151_JM_dhydA2 originated from pMTL84151[53] where the Gram-positive replicon was partially deleted. For deletion of the *hydA2* gene (Awo_c08260) by homologous recombination, each 500 bp of upstream and downstream flanking regions (UFR and DFR) of *hydA2* (Awo_c08730) were inserted into the multiple cloning sites of the plasmid. As selection marker, the plasmid has a *catP* gene from *Clostridium perfringens* for chloramphenicol/thiamphenicol resistance[54] and a *pyrE* gene from *Eubacterium limosum* for counter selection against 5-fluoroorotic acid[31]. After transformation of pMTL84151_JM_dhydA2 into the Δ*hydBA* mutant by electroporation (625 V, 25 μF, 600 Ω, in 1 mm cuvettes), the integrants were selected on an agar plate with bicarbonate-buffered complex medium containing 20 mM fructose, 50 mM formate and 30 ng/μl thiamphenicol. Subsequently, disintegration was carried out on an agar plate with bicarbonate-buffered minimal medium[35] containing 20 mM fructose, 50 mM formate, 1 μg/mL uracil and 1 mg/mL 5-fluoroorotic acid. The deletion of the *hydA2* gene was analyzed by PCR with primers which bind upstream of UFR and downstream of DFR: aus_hydA2_for (5′-GGCGAACAGATGAATGTTTATACTC-3′) and aus_hydA2_rev (5′-GCAGGTCGTTTCACCAACTA-3′). To confirm the purity of the mutant, primers binding inside of the *hydA2* genes were used: in_hydA2_for (5′-AGAAATGCAAATCCTATGTTTCCATC-3′) and in_hydA2_rev (5′-CACCTAGCGGTTGATCGAA-3′). The deleted region of the mutant was further verified by Sanger sequencing[55].

### SNP analysis

The genomes of the CO-adapted Δ*hydBA/hydA2* mutant were analyzed for SNP[56]. Genomic DNA of the Δ*hydBA/hydA2* mutant was isolated using the DNeasy Blood & Tissue Kit (Qiagen, Hilden, Germany) following the manufacturer's instruction. Sequencing was carried out on a MiSeq system with the reagent kit v3 with 600 cycles (Illumina, San Diego, CA, USA) following the manufacturer's instruction. For SNP analysis, Illumina raw reads were trimmed with trimmomatic v.0.39 and mapped against the reference genome of *A. woodii*[25] using bowtie2 v.2.4.1 with end-to-end mode. The pileup file for SNP calling was generated with samtools v.1.9 and subsequently the detection of SNPs was performed with Breseq v.0.32.0.

### Preparation of resting cells

Cells of *A. woodii* were cultivated either with 20 mM fructose, or CO (50%, 2 bar, headspace of 800 mL) in 1 L bicarbonate-buffered complex medium under a $N_2/CO_2$ atmosphere (80:20, v/v) to the late exponential growth phase (with 20 mM fructose, $OD_{600}$ of 1.5; with 50% CO for the Δ*hydBA/hydA2* mutant, $OD_{600}$ of 0.8). For conversion of formate, cells were grown in 1 L phosphate-buffered complex medium under a 100%

$N_2$ atmosphere with 100 mM formate to the late exponential growth phase (Δ*pyrE*, $OD_{600}$ of 0.3; Δ*hydBA/hydA2*, $OD_{600}$ of 0.8). Cells were pelleted by centrifugation (Avanti J-25 and JA-10 Fixed-Angle Rotor; Beckman Colter, Brea, CA, United States) at 11,295 × g and 4 °C for 10 min, followed by washing with 30 mL of imidazole buffer (50 mM imidazole, 20 mM KCl, 20 mM $MgSO_4$, 4 mM DTE, 4 μM resazurin, pH 7.0) by centrifugation at 8716 × g and 4 °C for 10 min (Avanti J-25 and JA-25.50 Fixed-Angle Rotor; Beckman Colter, Brea, CA, United States). The cell pellets were resuspended in 5 mL imidazole buffer and kept in 16-mL Hungate tubes. All steps were performed under strictly anoxic conditions in an anoxic chamber (Coy Laboratory Products, Grass Lake, MI, United States) filled with $N_2/H_2$ (96-98%/2-4%; v/v). The residual $H_2$ in the cell suspensions was removed by changing gas phase to 100% $N_2$. For the determination of total protein concentration, 1 mL of 1:100 diluted cell suspensions were mixed with 125 μL of 4 M NaOH and incubated at 100 °C for 10 min[57]. Subsequently, the mixture was placed on ice for 2 min, mixed with 400 μL reagent (60 mM K-Na-tartrate, 250 mM NaOH, 10 mM $CuSO_4$, 38 mM KI), and incubated at 37 °C for 30 min. After centrifugation at 13300 rpm for 5 min, extinction from 1 ml supernatants was determined at a wavelength of 546 nm. 0 to 1 mg/mL bovine serum albumin (BSA) was used for calibration.

### Cell suspension experiments

For conversion of CO, resting cells were prepared in 10 mL of imidazole buffer (50 mM imidazole, 20 mM KCl, 20 mM NaCl, 20 mM $MgSO_4$, 4 mM DTE, 4 μM resazurin, pH 7.0) in 120 mL serum flasks to a final protein concentration of 1 mg/mL. For the experiments under $CO_2$/bicarbonate-depleted conditions, a $N_2/CO$ gas phase (50:50, v/v; headspace of 110 mL) was used. When adding 60 mM or 300 mM $KHCO_3$, the headspace of the cell suspensions was changed to a $N_2/CO_2/CO$ atmosphere (40:10:50, v/v/v; headspace of 110 mL). For the experiments under $Na^+$-depleted conditions, $Na^+$-depleted buffer (50 mM imidazole, 20 mM KCl, 20 mM $MgSO_4$, 60 mM $KHCO_3$, 4 mM DTE, 4 μM resazurin, pH 7.0) was used and the contaminating $Na^+$ concentration in the buffer was measured with an Orion sodium electrode (Thermo Fischer Scientific, Waltham, MA, USA) following the manufacturer's protocol. For conversion of formate, resting cells were prepared in 10 mL of bicarbonate-containing imidazole buffer (50 mM imidazole, 20 mM KCl, 20 mM NaCl, 20 mM $MgSO_4$, 20 mM $KHCO_3$, 4 mM DTE, 4 μM resazurin, pH 7.0) under a $N_2/CO_2$ atmosphere (80:20, v/v; headspace of 110 mL) to a final protein concentration of 1 mg/mL and 250 mM sodium formate was added as energy source. Resting cells were incubated at 30 °C in water bath with shaking (150 rpm) and 1-mL samples were collected for determination of metabolites or ATP contents.

### Determination of ATP concentration in resting cells

To prepare samples for ATP measurement, 400 μL of the cell suspensions was mixed with 150 μL of 3 M perchloric acid and kept on ice for 10 min. Subsequently, the mixtures were neutralized by adding 40 μL of saturated $K_2CO_3$ solution and 80 μL of 400 mM TES buffer (pH 7.6), and supernatants were collected after centrifugation at 12,000 × g and 4 °C for 5 min. For the determination of the ATP concentration, 100 μL of the supernatant was mixed with 50 μL of luciferase reagent (ATP Bioluminescence Assay kit CLS II, Roche, Basel, Switzerland) and the ATP-dependent light emission was measured at a wavelength of 560 nm in a microplate reader (FLUOstar Omega, BMG Labtech, Ortenburg, Germany). A calibration curve (0–500 pmol ATP/assay) was used for quantification.

### Metabolite analyzes

The concentrations of fructose, formate, acetate, and lactate were determined by high-performance liquid chromatography[58] and Chromeleon v.6.8 software was used for analyzing data. $H_2$, CO or ethanol were determined using gas chromatography[22,59] and TotalChrom Navigator v.6.3.2.0646 software was used for analyzing data.

## Preparation of cell-free extracts

To generate cell-free extracts, *A. woodii* was grown on 20 mM fructose (Δ*pyrE* and Δ*hydBA/hydA2*) or 20 mM fructose plus 50 mM formate (Δ*hydBA/hdcr*) in 500 mL bicarbonate-buffered complex media under a $N_2/CO_2$ atmosphere (80:20, v/v; headspace of 500 mL). At the late exponential growth phase ($OD_{600}$ of 1.0), cells were harvested by centrifugation (Avanti J-25 and JA-10 Fixed-Angle Rotor; Beckman Colter, Brea, CA, United States) at 11,295;× *g* and 4 °C for 10 min and washed in 30 mL lysis buffer (25 mM Tris-HCl, 20 mM $MgSO_4$, 2 mM DTE, and 4 µM resazurin, pH 7.5) by centrifugation at 8719 × *g* and 4 °C for 10 min (Avanti J-25 and JA-25.50 Fixed-Angle Rotor; Beckman Colter, Brea, CA, United States). Then, cell pellets were resuspended in 2.5 mL lysis buffer containing 100 mM PMSF and a spatula tip of DNase I, and cell suspensions were disrupted by two passages through a French press (900 Psi). Subsequently, cell-free extracts were separated from cell debris and whole cells by centrifugation (Microcentrifuge SD, Carl Roth, Karlsruhe, Germany). All steps were carried out under strictly anoxic conditions in an anoxic chamber (Coy Laboratory Products, Grass Lake, MI, United States) filled with $N_2/H_2$ (96–98%/2–4%; v/v). The residual $H_2$ in the cell-free extracts was removed by changing the gas phase to 100% $N_2$. The protein concentration was determined by the Bradford assay[60].

## Enzymatic assays

Formate-dependent reduction of methyl viologen was analyzed in 1 mL assay buffer (100 mM HEPES, 20 mM $MgSO_4$, 2 mM DTE, and 4 µM resazurin, pH 7.0) in 1.8 mL anoxic cuvettes (Glasgerätebau Ochs, Bovenden, Germany) under a $N_2$ atmosphere at 30 °C[26]. First, 1 mM methyl viologen and 50 µg cell-free extract were added to the assay. The reaction started by adding 10 mM formate, and the reduction of methyl viologen was monitored at 604 nm. Formate-dependent Fd reduction was carried out in 1 mL assay buffer in 1.8 mL anoxic cuvettes under a $N_2$ atmosphere at 30 °C[26]. For measurements, 6 µM Fd prepared from *Clostridium pasteurianum*[61] and 1.0 mg cell-free extract were added to the assay. The reaction was started by adding 40 mM formate, and the reduction of Fd was followed at 430 nm.

Formate production from $Fd^{2-}$ and $CO_2$ was determined in 60 mL serum flasks filled with 2.5 mL assay buffer under a $N_2/CO_2/CO$ atmosphere (72:18:10, v/v/v) at 30 °C in a shaking water bath[37]. 100 µg CODH purified from *A. woodii*[8] and 3.0 mg cell-free extract was added to the assay. The reaction was started by adding 6 µM Fd. Formate production from $H_2$ plus $CO_2$ was analyzed in 60 mL serum flasks filled with 2.5 mL assay buffer and 3.0 mg cell-free extract under a $H_2/CO_2$ atmosphere (80:20, v/v) at 30 °C in a shaking water bath[26]. Formate concentration was determined with an enzymatic assay kit (R-Biopharm, Darmstadt, Germany) following the supplier's protocol.

Hydrogen evolution from formate was analyzed in 6 mL serum flasks filled with 1 mL assay buffer and 3.0 mg cell-free extract under a 100% $N_2$ atmosphere[26]. The reaction was started by adding 40 mM formate. Hydrogen concentration was measured as described in the metabolite analyzes section.

## Reporting summary

Further information on research design is available in the Nature Portfolio Reporting Summary linked to this article.

## Data availability

The SNP-sequencing generated during this study is available in the Sequence Read Archive (SRA) with following accession codes: SRR28714882, SRR28714883 and SRR28714884. Source data are provided in this paper.

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

## Acknowledgements

V.M. is indebted to the European Research Council (ERC) for financial support under the European Union's Horizon 2020 research and innovation program (ACETOGENS, grant agreement no. 741791).

## Author contributions

V.M. designed and supervised the research analyzed the data and wrote the manuscript. J.M. designed the research, performed the experiments, analyzed the data, and wrote the manuscript. A.P. and R.D. performed SNP sequencing. The manuscript was approved by all authors.

## Funding

## Competing interests

The authors declare no competing interest.
