## [Peer Review File · Nature Communications]

Redirecting electron flow in *Acetobacterium woodii* enables growth on CO and improves growth on formateReviewers' Comments:

Reviewer #1:

Remarks to the Author:

Summary

The authors established a CO and formate bioconversion system using an *A. woodii* double knockout strain of HydA2/HydBA. Authors suggested that this strain exhibits increased tolerance to CO gas, although it cannot utilize H₂. The HDCR of *A. woodii* is known to be a CO-sensitive enzyme complex. However, based on the results described in this study, it is difficult to understand that hydrogenase deletion is a major factor for enabling the growth of *A. woodii* under CO gas conditions. In particular, the authors exposed the mutant strain to CO gas condition for six months for adaptation, but they did not confirm any genomic changes of this strain. Although the increased formate production is attractive, in acetogenic microorganisms, higher formate levels indicate a functional limitation of the methyl-branch, which leads to a decrease in ATP yield. Thus, several concerns are listed below for clarifying the sustainability of the system.

- 1) Explain the difference in CO sensitivity between *A. woodii* and other acetogens, for example *C. ljungdahlii* or *E. limosum*, and elaborate on why the deletion of hydrogenases in *A. woodii* was considered to enhance CO resistance in introduction.
- 2) Indicating the growth curve and metabolite production results for the wild type or Δ pyrE strain under CO conditions simultaneously would help understand the extent of growth inhibition by *A. woodii* under CO conditions.
- 3) In Line 113~114, on page 6, and Figure 2, the strain adapted under CO conditions exhibits consistent growth profiling regardless of CO concentration in Figure 2b. Despite an increase in the carbon source, CO, there seems to be no change in cell mass. This makes it challenging to consider CO utilization. However, the authors suggested that CO is utilized as a carbon source. If so, please present the results of CO gas consumption measurements.
- 4) In lines 106-107 on page 5, the authors mentioned that they deleted hydrogenases to enhance resistance to CO, but the optical density (O.D) was only 0.2. Showing how much the control strain's O.D increases under the same conditions would help understand the impact of hydrogenase deletion on CO tolerance.
- 5) In lines 106-107, on page 5, despite the deletion of hydrogenases, there was still growth inhibition observed under 25% CO conditions. Therefore, the authors explained that they exposed the strain to long-term CO conditions for adaptation. This interpretation suggests that, ultimately, hydrogenase deletion did not have a significant impact on CO.
- 6) In lines 106-107, on page 5, Please provide a detailed explanation of how adaptation was carried out under CO gas conditions.
- 7) In lines 105-110, on page 5, under the same 25% CO gas conditions, the adapted strain showed an increase in growth rate and O.D. But this strain produced same amount of acetate. It appears necessary to confirm whether CO gas or CO₂ gas was more predominantly utilized under this condition.
- 8) In line 115~117, on page 6, Given the previous results, it seems that adaptation under CO conditions has contributed more to CO resistance than hydrogenase deletion. In this case, it appears necessary to identify any mutations that occurred in the genome and to provide an explanation for the mutations that the authors claim influenced the lag-phase growth.
- 9) In line 120~132, on page 6, A quantitative comparison of the precise amount of gas introduced into the headspace and the conversion yield to formate or acetate seems necessary.
- 10) In line 140~158, on page 7~8, on section; 'Fd-dependent FDH activity in the CO-adapted Δ hydBA/hydA2 mutant', To facilitate a clearer understanding of the authors' explain, it might be beneficial to present the information in a figure or visual representation.
- 11) In line 177~178, on page 8, To substantiate the suggestion, data comparing ATP yield would be essential.
- 12) On figure 4, 5, It appears that the control strain (Δ pyrE) utilizes formate more effectively, requiring an explanation. Additionally, the rationale for utilizing the hydrogenase deletion strain seems insufficient. While the max OD is higher in the adapted strain, the growth rate appears to be similar.

However, the formate utilization in the control strain seems superior to the hydrogenase mutant strain. This aspect is not stoichiometric understood.

13) On discussion parts, to provide evidence for Figure 6, experiments involving enzyme purification were conducted. Since the content of these experiments is crucial for understanding redox balancing in the KO strain, it is recommended to create a new section rather than including them in the discussion.

I write several minor review comments as below;

- 1) In all Figure and Supplementary Figure, please show all biological replicates and include error bars, over n=3
- 2) In line 227, on page 11, the term "Rnf/hydrogenase complex" could be read as if Rnf complex and hydrogenase form a complex together. It would be more accurate to express it as "Rnf or hydrogenase complex" for a clearer understanding.
- 3) On method sections, please explain the headspace volume of each cultivation to understand the stoichiometry of CO/CO₂ to formate or acetate.
- 4) In line 287, on page 13, for clarity, please express "DNA sequencing" as "Sanger sequencing" to differentiate the technique.
- 5) In line 279, 281, 290, 293, on page 13, for clarity, please represent "liter" as "L" and "milliliter" as "mL" to differentiate the units.
- 6) In line 297, 299 on page 14, for precise expression, please represent "rpm" units as "x g" to indicate the relative centrifugal force.
- 7) In line 297, 301, 308, 310 on page 14, for clarity, please represent "liter" as "L" and "milliliter" as "mL" to differentiate the units.
- 8) In line 317, 320, 321, 325, 327, 329 on page 15, for clarity, please represent "liter" as "L" and "milliliter" as "mL" to differentiate the units.
- 9) In line 328, on page 15, for precise expression, please represent "rpm" units as "x g" to indicate the relative centrifugal force.
- 10) On method of measurement of ATP contents, Please provide the wavelength of bioluminescence used for ATP measurements.
- 11) In line 341, on page 16, for accuracy, please use "with" or "plus" instead of '+'.
- 12) In line 344, 345 on page 16, for precise expression, please represent "rpm" units as "x g" to indicate the relative centrifugal force.
- 13) In line 344, 347, 357, 358, on page 16, for clarity, please represent "liter" as "L" and "milliliter" as "mL" to differentiate the units.
- 14) In line 362, 366, 369, 370, 373, on page 17, for clarity, please represent "liter" as "L" and "milliliter" as "mL" to differentiate the units.
- 15) In line 369, on page 17, for accuracy, please use "with" or "plus" instead of '+'.
- 16) In line 551, in figure 3 legends, Please accurately describe the gas compositions in both (a) and (b). Currently, it seems that the gas compositions placed in (a) and (b) are perceived as identical.

Reviewer #2:

Remarks to the Author:

Overall: The overall strength of the paper is that the authors reveal a seemingly straightforward and reproducible mechanism that *A. woodii* can recover from inhibition by CO. The major weakness is that the mechanism and the metabolic consequences are not clear. The authors indicate that the work is important for biotechnological applications, but the experiments are performed at a small scale at low substrate concentrations and without mass balance analyses. There also appears to be some uncoupling of CO metabolism with respect to product formation. Ethanol was mentioned but only analyzed with respect to growth on fructose. Thus, with respect to bioconversion, this process is ambiguous. The results with *A. woodii* seem to be similar to those described to some performed in *A. kivui*. The link of the results of the role of HDCR and its relationship to ferredoxin and CODH is also rather unclear. Therefore, overall I feel that the results/conclusions in this paper are incomplete and

incremental. The paper also seems targeted to a narrow audience of people interested in acetogenic metabolism and is not written to be understood by the broader audience represented by Nature Communications.

1. Title: The authors should state "Bioconversion ..." into what? Cell mass, some product???
2. Abstract: Given that this is targeted to a general audience, some statement of significance of the results (to a general audience) is needed. Currently, the abstract seems targeted to a specialized journal. In addition, this statement, "... Δ HydA2 variant of the hydrogen-dependent CO₂ reductase" is unclear without an at-least brief description of what is the "CO₂ reductase". In the discussion it is stated: "Chemical and biological methods to produce formate from H₂ + CO₂ are available^{23,36-39} and may then be used by the Δ hydBA/hydA2 mutant to increase yields of added-value compounds such as acetone dramatically. This is a big step in overcoming energetic barriers in acetogenic one-carbon conversion." It really isn't clear what "This" is. Perhaps they are talking about the results of this paper?
3. Page 3, Line 44: "CO/CO₂ allows for the reduction of the electron acceptor ferredoxin (E_{0'} ≈ -500 mV)." This statement should be cited and it should be noted that ferredoxins (even those in acetogens) have a wide range of redox potentials. Do the authors know the redox potential for the ferredoxin involved in this CO₂ reductase reaction?
4. Page 3: "In the model acetogenic, Rnf-containing bacterium *Acetobacterium woodii*, it is calculated to 1.5 ATP/acetate compared to only 0.3 mol ATP/acetate when grown on H₂ and CO₂." Please supply the reference.
5. Page 3: "In this statement, "the only industrial application that uses acetogenic bacteria to date is the production of ethanol from syngas, a gas mixture that contains CO as well as CO₂ and H₂", the authors cite ref 16, which appears to refute the statement. 16. Liew, F.E. et al. Carbon-negative production of acetone and isopropanol by gas fermentation at industrial pilot scale. *Nat Biotechnol* 40, 335-344 (2022), which states: "... we scaled-up our optimized strains for continuous production [of acetone and isopropanol] at rates of up to ~3 g/L/h and ~90% selectivity.
6. Page 4 lines 71-72. "A FeFe-hydrogenase module (HydA2) and a formate dehydrogenase module (FDH) sit on the wire like bulbs in a fairy light". This is a poetic but not very descriptive sentence. Perhaps the bulbs are the redox centers? What do they mean by a "fairy light"? Please describe this in terms that the audience can picture the arrangement of the redox centers (or the proteins).
7. P. 5. Line 85: "... these cells also grew to much higher cell yields ..." Please state, much higher cell yields than what?
8. Results, P. 5. Line 97 & Suppl. Fig. 2: " Δ hydBA/hdcr mutant (ref 28), the Δ hydBA/hydA2 mutant could grow on fructose alone with growth rate and final OD₆₀₀ similar to the Δ pyrE strain". The data with the Δ hydBA/hdcr mutant is also shown in Suppl. Fig. 2. How does this growth compare to that of the WT strain? Why was the Δ pyrE strain chosen for comparison?
9. Results, Suppl. Fig. 2. "The growth experiments were performed in biological triplicates and one representative growth curve is presented. How representative is this curve? The data from all three replicates (or the average OD with standard deviation) should be presented.
10. Results, The presentation of a single representative data set throughout the paper (Suppl. Fig. 2, Fig. 2, Fig 3, is inadvisable and not acceptable in most journals, some of which require all three data sets and others allow an average with a standard deviation, as the authors showed in Figs 4 and 5 and in Suppl Fig. 3.

11. Results, Figure 1 needs extensive revision. As drawn, it implies that ACS (the subunit of the CODH/ACS complex), not CODH/ACS, reduces CO₂. That is incorrect. The ACS subunit does not catalyze CO₂ reduction or CO oxidation. That function is supplied by the CODH component of the complex. Unlike what is presented in this figure, that CODH component is capable of reversible oxidation of CO to CO₂ (a function that can also be supplied by a free-standing CODH, which is apparently what they are indicating for CooS). It also isn't clear why HDCR, which has HydA2 as a component subunit are both shown in the reduction of CO₂ to formate and why this protein is depicted with blue, green, and purple circles. Furthermore, it is ACS that reacts with CO (why [CO]?) the methylated CFeSP (not shown, and not methyl-THF) and CoA (not shown) to make acetyl-CoA. Furthermore, the coupling of RNF with the reactions on the left are not indicated. "the CO dehydrogenase/acetyl-CoA synthase/ is generally referred to as CODH/ACS, not "Acs" alone."

12. Results, P. 5 Line 108: "could grow on 25% CO to a final OD₆₀₀ of 0.4 with a growth rate of 0.02 h⁻¹ (Fig. 2a)." Please compare to the data on H₂/CO₂ for wt proteins on Fructose, H₂/CO₂, and on CO.

13. Results, Line 112, "Apparently, the mutant not only grew with 25% CO, but... " Why apparently? Is there a questions about whether or not there was growth?

14. Results, Page 6, Line 114: "After growth, similar amounts of acetate (21 to 25 mM) were produced, regardless of the CO concentrations" How about the mass balance?

15. Results, Line 138. "acetogenesis from CO also requires Na⁺/the Rnf complex". This seems important enough that the authors might wish to highlight in the abstract.

16. Results, Line 169. "After ten transfers, the [Δ hydBA/hydA2] mutant grew on 100 mM formate with a similar growth rate as the Δ pyrE strain but produced twice the amount of biomass (final OD₆₀₀ of 0.9) (Fig. 4a)." Did the level of hdcr increase? What is responsible for the higher OD?

17. Results. As far as I can tell, there is no information presented here that Fd is the redox mediator or which ferredoxin is involved in the hbcR reaction. If there is a hbcR isolated with the complex, has this been deleted? This is problematic throughout; for example, in the discussion, they make a number of apparent assumptions about Fd - i.e., Line 235 - ... indicating again that not CO toxicity but HDCR adaptation to ferredoxin".

18. Results. Section on conversion of formate in resting cells of the CO-adapted Δ hydBA/hydA2 mutant. It seems problematic that the rates of formate conversion do not match that of acetate formation. The authors should explain this. How about the rate of CO₂ formation? CO formation?

19. Discussion. Line 214. The CO metabolism section should be more explicit. "Carbon monoxide metabolism in the Δ hydBA/hydA2 mutant is postulated to involve the HDCR." Here they need a reference if it has been proposed below. Perhaps this is new. "In this case, reduced ferredoxin ..." Perhaps they mean CO dependent reduction of ferredoxin ..."

20. "In stark contrast, in the Δ hydBA/hydA2 mutant formate is oxidized with simultaneous reduction of ferredoxin (Fig. 6d)." However, Fig 6 doesn't present results, but a scenario.

21. Growth on CO seems to lead predominantly to formate, not acetate. It seems possible that the major effect of the adaptation is relief from CO inhibition. Perhaps CO is converted to either CO₂ or formate, which are used for acetate (+?) formation.

22. As shown in Suppl. Fig. 2, ethanol is also produced. it also seems that ethanol as well as acetate concentrations should have been determined during all the fermentations (especially on CO and formate) and it would be helpful to see a detailed mass/balance related to CO, CO₂, acetate, ethanol

and formate during the experiments. Also in a related issue, what happens during growth on CO + formate? Is formate preferentially utilized? Is the CO converted to formate which equilibrates with CO₂ as the substrate for acetogenesis.

Reviewer #3:

Remarks to the Author:

The manuscript of Moon and Mueller describes the development and characterization of a hydrogenase-free (with knockout of the two hydrogenases HydA2 and HydB) and CO-adapted mutant of the acetogenic bacterium *Acetobacterium woodii*. Unlike the wildtype, the mutant can grow on CO and formate and convert them to produce acetate and ATP. The hydrogenase-free Δ hydBA/hydA2 mutant can directly use CO to generate formate and reduced ferredoxin as electron donor instead of H₂. The reduced ferredoxin can further fuel the respiratory enzymes ferredoxin:NAD-oxidoreductase (Rnf) complex that couples electron transfer to Na⁺-dependent ATP generation. Conversion of formate in resting cells of the CO-adapted Δ hydBA/hydA2 mutant was quantitatively studied and compared to that of a Δ pyrE strain. This is an interesting work with significant advance regarding conversion of CO and formate by *A. woodii*.

However, there are some major issues that need to be addressed for further improving the quality of the work before publication.

- 1) In general, regarding the double deletion mutant which was adapted to grow on CO, the authors should perform genome re-sequencing to identify other possible mutations occurred during adaptation.
- 2) The roles of the other mutations should be clarified or at least discussed regarding their possible roles in affecting use of CO and formate and generation of energy (ATP).
- 3) L27, the Δ HydA2 variant is not described in the manuscript. The 25-fold formate production is achieved by Δ HydBA/HydA2. More information about Δ HydBA/HydA2 should be added.
- 4) Line 50, "the calculated 1.5 ATP/acetate" which is shown in figure 6a, needs to be explained more clearly. And add reference(s).
- 5) L124, it is uncertain whether the additional bicarbonate will be decomposed into CO₂ and utilized as a carbon source? Why is the buffer different (bicarbonate or phosphate) when utilizing CO or formate?
- 6) L133, why formate is also produced in the absence of bicarbonate? Pl give an explanation for this.
- 7) L527, why three quarters of formate are oxidized? Is it achieved from your calculation or reference? Is the oxidization ratio the same when using Fd₂- instead of H₂ as electron carrier as shown in figure.6d?
- 8) L101, "a reductant other than H₂".
- 9) L134, about Fig.3c, it should be explained with more details.
- 10) L258, information about the Δ pyrE strain is missing and should be added.

Point-by-point response to the reviewer's comments.

Reviewer #1 (Remarks to the Author):

Summary

The authors established a CO and formate bioconversion system using an *A. woodii* double knockout strain of HydA2/HydBA. Authors suggested that this strain exhibits increased tolerance to CO gas, although it cannot utilize H₂. The HDCR of *A. woodii* is known to be a CO-sensitive enzyme complex. However, based on the results described in this study, it is difficult to understand that hydrogenase deletion is a major factor for enabling the growth of *A. woodii* under CO gas conditions. In particular, the authors exposed the mutant strain to CO gas condition for six months for adaptation, but they did not confirm any genomic changes of this strain. Although the increased formate production is attractive, in acetogenic microorganisms, higher formate levels indicate a functional limitation of the methyl-branch, which leads to a decrease in ATP yield. Thus, several concerns are listed below for clarifying the sustainability of the system.

Answer: Sorry that we did not make this point more clear. You have specified your general remarks above in your specific comments that we addressed below. Here, we would like to make only a few general comments: First, without deletion of the hydrogenases *A. woodii* could not be adapted to grow on CO. Therefore, hydrogenases are a major factor. Second, yes there is another factor that changes within the ca. 6-month adaptation period and as suggested, we have performed genome (SNP) analyses that are included in the revised version. Third, we fully agree that an accumulation of formate could result from a shortage of ATP. However, this was NOT observed in growing cells, only in resting cells. Whether ATP shortage is true or not, has not been addressed in detail since this is only a side aspect of our study and not relevant for the conclusions.

1) Explain the difference in CO sensitivity between *A. woodii* and other acetogens, for example *C. ljungdahlii* or *E. limosum*, and elaborate on why the deletion of hydrogenases in *A. woodii* was considered to enhance CO resistance in introduction.

Answer: Thank you for the comment. The difference in CO sensitivity in *C. ljungdahlii*, *E. limosum*, *T. kivui* and *A. woodii* is now discussed in the "Discussion", shortly, since we are space limited. Why we considered the deletion of hydrogenases in *A. woodii* to enhance CO resistance was already mentioned in the introduction in the first version. In short, FeFe hydrogenases are generally CO sensitive. *A. woodii* has only two hydrogenases, both are of the FeFe-type, and both have been shown experimentally to be inhibited by CO.

2) Indicating the growth curve and metabolite production results for the wild type or Δ pyrE strain under CO conditions simultaneously would help understand the extent of growth inhibition by *A. woodii* under CO conditions.

Answer: Thank you for the comment. The growth curve of the Δ pyrE strain is shown in the revised manuscript in Fig. 2a. Since the Δ pyrE strain did not grow on CO, metabolites were not produced.

3) In Line 113~114, on page 6, and Figure 2, the strain adapted under CO conditions exhibits consistent growth profiling regardless of CO concentration in Figure 2b. Despite an increase in the carbon source,

CO, there seems to be no change in cell mass. This makes it challenging to consider CO utilization. However, the authors suggested that CO is utilized as a carbon source. If so, please present the results of CO gas consumption measurements.

Answer: Thank you for pointing this important point out. Cells clearly grew on CO, there was no growth when CO was omitted. In the revised version we added the values for the optical densities of the culture incubated without CO to Fig. 2b. The fact that the final yields did not increase when already high CO concentration of 25% was further increased up to 100% indicates that CO was not the limiting factor at high CO concentrations. To make this more clear we changed the terminology to “CO tolerance” at high CO concentrations. CO gas consumption measurements have been done with resting cells (no CO incorporation into biomass) and are presented in the revised version.

4) In lines 106-107 on page 5, the authors mentioned that they deleted hydrogenases to enhance resistance to CO, but the optical density (O.D) was only 0.2. Showing how much the control strain's O.D increases under the same conditions would help understand the impact of hydrogenase deletion on CO tolerance.

Answer: Thank you for the comment. As mentioned in the question 2), the control strain $\Delta pyrE$ does not grow on CO. The values for the optical densities of the control strain were added to Fig. 2a in the revised manuscript.

5) In lines 106-107, on page 5, despite the deletion of hydrogenases, there was still growth inhibition observed under 25% CO conditions. Therefore, the authors explained that they exposed the strain to long-term CO conditions for adaptation. This interpretation suggests that, ultimately, hydrogenase deletion did not have a significant impact on CO.

Answer: As mentioned above and in the manuscript, we were not able to adapt the wild type or the $\Delta pyrE$ strain to grow on CO. This illustrates the importance to delete the CO-sensitive hydrogenases. See also Bertsch and Müller, 2015, where we reported that the wild type does not grow on CO but on CO + formate, a growth substrate combination that does not require hydrogenase activity. Therefore, the initial deletion of the hydrogenases was the key to get CO-dependent growth at the first place. The further increase in cell yields over time is a fine tuning of metabolism seen in almost every bacterium that gets adapted to a new growth substrate. For example, growth of the acetogen *Sporomusa* on methanol can be optimized by ALE, same for growth of *Thermococcus onnurineus* on formate, just to mention two recent examples from the same physiological group.

6) In lines 106-107, on page 5, Please provide a detailed explanation of how adaptation was carried out under CO gas conditions.

Answer: By serial transfer of stationary cultures to culture vessels with the next higher CO concentration. Details are included in the revised version of the manuscript.

7) In lines 105-110, on page 5, under the same 25% CO gas conditions, the adapted strain showed an increase in growth rate and O.D. But this strain produced same amount of acetate. It appears necessary to confirm whether CO gas or CO₂ gas was more predominantly utilized under this condition.

Answer: We were probably not clear on this point which led to a misunderstanding. In lines 105-110 we describe only the optical densities during the adaptation process, not product formation. When the culture was finally adapted to CO, we grew it at different CO concentrations, starting at 25% CO. As already discussed above, higher CO concentrations were tolerated but the culture was not CO limited and concentrations >25% led to the same final yield and acetate concentration. This has been made clearer in the revised version (see above, comment #3).

8) In line 115~117, on page 6, Given the previous results, it seems that adaptation under CO conditions has contributed more to CO resistance than hydrogenase deletion. In this case, it appears necessary to identify any mutations that occurred in the genome and to provide an explanation for the mutations that the authors claim influenced the lag-phase growth.

Answer: Yes, we fully agree. We have sequenced the genome before and after adaptation. The data have been added to the revised version of the manuscript in the new paragraph “Genetic mutations in the genome of the $\Delta hydA2/hydBA$ mutant during adaptation on CO”. A table describing the SNPs has been added as well, and in the discussion, we now highlight the genetic changes determined in the light of genetic changes reported for other bacteria that have been adapted to grow on CO.

9) In line 120~132, on page 6, A quantitative comparison of the precise amount of gas introduced into the headspace and the conversion yield to formate or acetate seems necessary.

Answer: Here, the amount of CO consumed was not determined because the focus was not on the determination of a fermentation balance but qualitatively show the observed and surprising formate production under these conditions in resting cells. However, to address the point of the reviewer we have added CO consumption from start and endpoint determinations of CO to the revised version of the manuscript.

10) In line 140~158, on page 7~8, on section; ‘Fd-dependent FDH activity in the CO-adapted $\Delta hydBA/hydA2$ mutant’, To facilitate a clearer understanding of the authors' explain, it might be beneficial to present the information in a figure or visual representation.

Answer: Thank you for this comment. We thought about a solution for a visual presentation but could come up only with a cartoon which would be very scholarly. Instead, we have fine-tuned the text to explain better the different enzyme activities for a clearer understanding.

11) In line 177~178, on page 8, To substantiate the suggestion, data comparing ATP yield would be essential.

Answer: The statement was kept vague on purpose (*arguing that...*). But the reviewer is right, ATP measurements would strengthen the argument. However, the measurement of the ATP concentration in growing cells is very erroneous for several technical reasons and, therefore, this is usually done with resting cells. We also determined the ATP content with resting cells and presented the data already in the first version of the manuscript: the ATP content is increased by 70% (line 201-203 in the first version; line 224-226 in the revised version).

12) On figure 4, 5, It appears that the control strain ($\Delta pyrE$) utilizes formate more effectively, requiring an explanation. Additionally, the rationale for utilizing the hydrogenase deletion strain seems insufficient.

While the max OD is higher in the adapted strain, the growth rate appears to be similar. However, the formate utilization in the control strain seems superior to the hydrogenase mutant strain. This aspect is not stoichiometric understood.

Answer: The reviewer is right, and we agree that the stoichiometry is not understood. Even more, the data are not (yet) understood on a physiological/mechanistic level. Clear is, that the final yield (biomass production) is doubled, and we present a plausible bioenergetic model for this increase (see Fig. 6). We also present data that support our notion about an increased ATP synthesis by experimentally demonstrating that the ATP concentration is increased by 70% in the double mutant. Yes, resting cells of the double mutant oxidize formate slower than the $\Delta pyrE$ strain, but this is NOT reflected in the growth rate, indicating that the formate oxidation rate is not rate limiting. The decrease in formate oxidation rate can have several reasons. We would favour a disbalance in the redox pool due to higher rates of ferredoxin reduction/high concentrations of reduced ferredoxin; reoxidation of reduced ferredoxin by the Rnf complex generates a membrane potential that will thermodynamically inhibit further ferredoxin oxidation. This would slow down further formate oxidation. Another likely explanation is that the HDCR is not optimally adjusted to use reduced ferredoxin as electron donor and that this is the rate-limiting step. But this is purely speculative and, therefore, we would not like to include this in the text.

13) On discussion parts, to provide evidence for Figure 6, experiments involving enzyme purification were conducted. Since the content of these experiments is crucial for understanding redox balancing in the KO strain, it is recommended to create a new section rather than including them in the discussion.

Answer: We are a bit puzzled by this comment. We did not present experimental data from this study in the discussion but refer to literature data. For example, the only enzyme activity that was mentioned in the discussion is: "HDCR...purified.... with only 6% activity". This value is not from this study but the literature and the reference is given at the end of the sentence. We do not find more enzymatic data in the discussion, so it's not clear to us what the reviewer is referring to.

I write several minor review comments as below;

1) In all Figure and Supplementary Figure, please show all biological replicates and include error bars, over $n=3$

Answer: we modified all figures of growth curves to show mean and error bars. When we include all biological replicates, the figures become confusing and unclear.

2) In line 227, on page 11, the term "Rnf/hydrogenase complex" could be read as if Rnf complex and hydrogenase form a complex together. It would be more accurate to express it as "Rnf or hydrogenase complex" for a clearer understanding.

Answer: changed.

3) On method sections, please explain the headspace volume of each cultivation to understand the stoichiometry of CO/CO₂ to formate or acetate.

Answer: The headspace volume was added for each cultivation in the method section.

4) In line 287, on page 13, for clarity, please express "DNA sequencing" as "Sanger sequencing" to differentiate the technique.

Answer: changed.

5) In line 279, 281, 290, 293, on page 13, for clarity, please represent "liter" as "L" and "milliliter" as "mL" to differentiate the units.

Answer: changed.

6) In line 297, 299 on page 14, for precise expression, please represent "rpm" units as "x g" to indicate the relative centrifugal force.

Answer: changed.

7) In line 297, 301, 308, 310 on page 14, for clarity, please represent "liter" as "L" and "milliliter" as "mL" to differentiate the units.

Answer: changed.

8) In line 317, 320, 321, 325, 327, 329 on page 15, for clarity, please represent "liter" as "L" and "milliliter" as "mL" to differentiate the units.

Answer: changed.

9) In line 328, on page 15, for precise expression, please represent "rpm" units as "x g" to indicate the relative centrifugal force.

Answer: changed.

10) On method of measurement of ATP contents, Please provide the wavelength of bioluminescence used for ATP measurements.

Answer: Wavelength used for ATP measurements is added in the revised manuscript.

11) In line 341, on page 16, for accuracy, please use "with" or "plus" instead of '+'.

Answer: changed.

12) In line 344, 345 on page 16, for precise expression, please represent "rpm" units as "x g" to indicate the relative centrifugal force.

Answer: changed.

13) In line 344, 347, 357, 358, on page 16, for clarity, please represent "liter" as "L" and "milliliter" as "mL" to differentiate the units.

Answer: changed.

14) In line 362, 366, 369, 370, 373, on page 17, for clarity, please represent "liter" as "L" and "milliliter" as "mL" to differentiate the units.

Answer: changed.

15) In line 369, on page 17, for accuracy, please use "with" or "plus" instead of '+'.

Answer: changed.

16) In line 551, in figure 3 legends, Please accurately describe the gas compositions in both (a) and (b). Currently, it seems that the gas compositions placed in (a) and (b) are perceived as identical.

Answer: Yes, the gas compositions in (a) and (b) are identical. However, we changed the sentence to be more accurate.

Reviewer #2 (Remarks to the Author):

Overall: The overall strength of the paper is that the authors reveal a seemingly straightforward and reproducible mechanism that *A. woodii* can recover from inhibition by CO. The major weakness is that the mechanism and the metabolic consequences are not clear.

Answer: Thank you for the positive comment, but of course, we disagree in part with the major weaknesses mentioned. The metabolic consequences are very clear to us: It is as simple as: *A. woodii* grows on CO! This is a consequence of deleting the only two hydrogenases in *A. woodii* as well as additional mutations that are presented in the revised version. Mechanistically, we propose that the HDCR can use ferredoxin as electron carrier under these conditions; that this is possible has been demonstrated before with mutants of *Thermoanaerobacter kivui*, but more important, by using purified HDCR complexes.

The authors indicate that the work is important for biotechnological applications, but the experiments are performed at a small scale at low substrate concentrations and without mass balance analyses.

Answer: Yes, of course, this is the first step, the creation of a strain of *A. woodii* that grows on CO. Scale up is important for any application, but this requires follow up studies with a completely different focus and different methodologies in fermenters and using systems biology approaches. Mass balance analyses are, of course, a must under scale up conditions in a fermenter. But for this story, we do not feel it to be essential to substantiate our conclusions. However, we added mass balance during cell suspension experiments to address the reviewers point.

There also appears to be some uncoupling of CO metabolism with respect to product formation.

Answer: Yes, but only in resting cells, and this is actually seen quite often, not only in acetogens. For example, we have published recently, that *A. woodii* can switch to hydrogenogenesis or mixed acid fermentation, but only in resting cells.

Ethanol was mentioned but only analyzed with respect to growth on fructose. Thus, with respect to bioconversion, this process is ambiguous.

Answer: *A. woodii* has 11 putative alcohol dehydrogenases, therefore, part of acetyl-CoA can be reduced to ethanol at the expense of 2 NADH. Ethanol production of *A. woodii* during growth on fructose has been already studied (Moon and Müller, 2021). Under Na⁺-depleted conditions where the WLP is impaired, cells produce ethanol from fructose. Likewise, due to the alteration of the WLP in the $\Delta hydBA/hydA2$ mutant, a minor amount of acetyl-CoA may be reduced to ethanol.

The results with *A. woodii* seem to be similar to those described to some performed in *A. kivui*.

Answer: Yes, the conclusion is that the HDCR present in both acetogens, *T. kivui* and *A. woodii*, can use ferredoxin as electron carrier under physiological conditions. However, the metabolic consequences are completely different. *T. kivui* has no Rnf, but Ech as respiratory enzyme, *A. woodii* has only Rnf. The Rnf complex allows for reoxidation of reduced ferredoxin coupled to ATP synthesis; the more reduced ferredoxin, the more ATP in *A. woodii*. This is completely different in *T. kivui* where the ferredoxin pool is linked *via* Ech to the hydrogen pool and electrons escape as hydrogen into the environment. In *A. woodii*, they all go into the respiratory chain!

The link of the results of the role of HDCR and its relationship to ferredoxin and CODH is also rather unclear.

Answer: We disagree on that point. It has been experimentally shown (and published) that the HDCR from *A. woodii* can use ferredoxin as electron carrier. Such data had also been included in this manuscript. I guess there is no doubt and overwhelming experimental evidence from many groups that the monofunctional CooS and the bifunctional CODH/ACS both can oxidize CO with concomitant reduction of ferredoxin.

Therefore, overall I feel that the results/conclusions in this paper are incomplete and incremental. The paper also seems targeted to a narrow audience of people interested in acetogenic metabolism and is not written to be understood by the broader audience represented by Nature Communications.

Answer: Thank you for the comment. We went through the manuscript again and feel that we had a good balance of being broad enough for a broader audience, but also specific enough for the specialized community. We note that none of the other reviewers made that point. Nevertheless, due to the addition of new data (SNP analyses) we had to re-write the abstract and feel that it is now better suited for a broader audience.

1. Title: The authors should state “Bioconversion ...” into what? Cell mass, some product???

Answer: Thanks, this is a good point. Here bioconversion means conversion of CO to acetate or formate and conversion of formate to acetate and higher cell mass. However, since the manuscript was transferred from Nature Biotechnology to Nature Communications, we had to reformat the

revised manuscript. This included the title which was too long and had to be shortened to fit to the 15-word-limit of the journal. The new title does not have the word “bioconversion” any more.

2. Abstract: Given that this is targeted to a general audience, some statement of significance of the results (to a general audience) is needed. Currently, the abstract seems targeted to a specialized journal. In addition, this statement, “... □HydA2 variant of the hydrogen-dependent CO₂ reductase” is unclear without an at-least brief description of what is the “CO₂ reductase”.

Answer: Thank you for pointing this out. Due to the addition of new data (SNP analyses) we had to re-write the abstract anyway and feel that the new version is better suited for a broader audience.

In the discussion it is stated: “Chemical and biological methods to produce formate from H₂ + CO₂ are available^{23,36-39} and may then be used by the ΔhydBA/hydA2 mutant to increase yields of added-value compounds such as acetone dramatically. This is a big step in overcoming energetic barriers in acetogenic one-carbon conversion.” It really isn’t clear what “This” is. Perhaps they are talking about the results of this paper?

Answer: Thank you for the comment. ‘This’ means the yield of a compound such as acetone but also the possibility to produce compounds that could hitherto not been made due to a negative ATP balance. The sentence was changed accordingly.

3. Page 3, Line 44: “CO/CO₂ allows for the reduction of the electron acceptor ferredoxin (E₀’≈-500 mV).” This statement should be cited and it should be noted that ferredoxins (even those in acetogens) have a wide range of redox potentials. Do the authors know the redox potential for the ferredoxin involved in this CO₂ reductase reaction?

Answer: Reference is now given and we also state the redox potential span of ferredoxins in acetogens. No, *A. woodii* has eleven potential ferredoxin encoding genes, none of the proteins has been characterized. Therefore, the ferredoxin used by the HDCR and its redox potential is unknown.

4. Page 3: “In the model acetogenic, Rnf-containing bacterium *Acetobacterium woodii*, it is calculated to 1.5 ATP/acetate compared to only 0.3 mol ATP/acetate when grown on H₂ and CO₂.” Please supply the reference.

Answer: Both values come from the metabolic scenarios as the one depicted in Fig. 6. The values used for the calculation (ions translocated /ATP; ions translocated per 2 electrons) have been experimentally determined or are calculated based on free energy changes of the redox reaction and the magnitude of the electrical potential across the cytoplasmic membrane. The latter two values are referenced, the values for ATP/acetate are derived from Fig. 6.

5. Page 3: “In this statement, “the only industrial application that uses acetogenic bacteria to date is the production of ethanol from syngas, a gas mixture that contains CO as well as CO₂ and H₂”, the authors cite ref 16, which appears to refute the statement. 16. Liew, F.E. et al. Carbon-negative production of acetone and isopropanol by gas fermentation at industrial pilot scale. *Nat Biotechnol* 40, 335-344 (2022), which states: “... we scaled-up our optimized strains for continuous production [of acetone and isopropanol] at rates of up to ~3 g/L/h and ~90% selectivity.

Answer: Thank you for the comment. Although this is not yet commercialized, we added isopropanol and acetone in the sentence.

6. Page 4 lines 71-72. “A FeFe-hydrogenase module (HydA2) and a formate dehydrogenase module (FDH) sit on the wire like bulbs in a fairy light”. This is a poetic but not very descriptive sentence. Perhaps the bulbs are the redox centers? What do they mean by a “fairy light”? Please describe this in terms that the audience can picture the arrangement of the redox centers (or the proteins).

Answer: We love this expression and like to keep it. The sentence reads: *This remarkable enzyme has a long chain of small, iron-sulfur containing proteins (HycB2 and HycB3) that make a long electron wire. With that, it is clear that the redox centers are in the wire. The enzymes sit on the wire like “light bulbs”. The whole assembly is then, according to my dictionary, a “fairy light” or a “chain of lights”. Since the term “fairy light” has been used before, we prefer this term. It’s poetic, yes, but reaches a broader audience.*

7. P. 5. Line 85: “... these cells also grew to much higher cell yields ...” Please state, much higher cell yields than what?

Answer: Thank you for the comment. We added ‘much higher cell yields than the wild type strain’.

8. Results, P. 5. Line 97 & Suppl. Fig. 2: “ Δ hydBA/hdcr mutant (ref 28), the Δ hydBA/hydA2 mutant could grow on fructose alone with growth rate and final OD600 similar to the Δ pyrE strain”. The data with the Δ hydBA/hdcr mutant is also shown in Supp. Fig. 2. How does this growth compare to that of the WT strain? Why was the Δ pyrE strain chosen for comparison?

Answer: Thank you for the comment. For the generation of a deletion mutant of *A. woodii*, we use the uracil auxotrophy strategy for which deletion of the *pyrE* gene is a prerequisite. The Δ pyrE strain is, thus, the parental strain for any mutant that we make/use. Therefore, we always have to compare the phenotype of our mutants to the Δ pyrE strain, not the wild type. The Δ pyrE strain and the wild type have the same growth behaviour under the conditions described in this study.

9. Results, Suppl. Fig. 2. “The growth experiments were performed in biological triplicates and one representative growth curve is presented. How representative is this curve? The data from all three replicates (or the average OD with standard deviation) should be presented.

Answer: Thank you for the comment. As Reviewer 1 also requested, we depicted now the average OD with standard deviation.

10. Results, The presentation of a single representative data set throughout the paper (Suppl. Fig. 2, Fig. 2, Fig 3, is inadvisable and not acceptable in most journals, some of which require all three data sets and others allow an average with a standard deviation, as the authors showed in Figs 4 and 5 and in Suppl Fig. 3.

Answer: Thank you for the comment. As Reviewer 1 also requested, we depicted now the average OD with standard deviation for all growth experiments.

11. Results, Figure 1 needs extensive revision. As drawn, it implies that ACS (the subunit of the CODH/ACS complex), not CODH/ACS, reduces CO₂. That is incorrect. The ACS subunit does not catalyze CO₂ reduction or CO oxidation. That function is supplied by the CODH component of the complex. Unlike what is presented in this figure, that CODH component is capable of reversible oxidation of CO to CO₂ (a function that can also be supplied by a free-standing CODH, which is apparently what they are indicating for CooS). It also isn't clear why HDCCR, which has HydA2 as a component subunit are both shown in the reduction of CO₂ to formate and why this protein is depicted with blue, green, and purple circles. Furthermore, it is ACS that reacts with CO (why [CO]?) the methylated CFeSP (not shown, and not methyl-THF) and CoA (not shown) to make acetyl-CoA. Furthermore, the coupling of RNF with the reactions on the left are not indicated. "the CO dehydrogenase/acetyl-CoA synthase/ is generally referred to as CODH/ACS, not "Acs" alone."

Answer: we fully agree, but the points the reviewer raises come from oversimplification of Fig. 1. to make it more accessible for a broad audience. We happily bring in more details and a new version of Fig. 1 is presented in the revised version. To the role of CooS and CODH/ACS: we follow the general assumption that CODH/ACS is for anabolic reactions and CooS for catabolic reactions. Rnf and the CO₂ reduction pathway are coupled via the soluble redox carriers ferredoxin and NAD that are both shown in the scheme.

12. Results, P. 5 Line 108: "could grow on 25% CO to a final OD₆₀₀ of 0.4 with a growth rate of 0.02 h⁻¹ (Fig. 2a)." Please compare to the data on H₂/CO₂ for wt proteins on Fructose, H₂/CO₂, and on CO.

Answer: Thank you for the comment. Comparison of the double mutant and the control strain ($\Delta pyrE$) on fructose and H₂+CO₂ was already described in the section 'Generation of the hydrogenase-free mutant $\Delta hydBA/hydA2$. The $\Delta pyrE$ did not grow on CO (Please, refer to the revised Fig. 2a).

13. Results, Line 112, "Apparently, the mutant not only grew with 25% CO, but..." Why apparently? Is there a questions about whether or not there was growth?

Answer: Thank you for the comment. We removed this expression in this revision.

14. Results, Page 6, Line 114: "After growth, similar amounts of acetate (21 to 25 mM) were produced, regardless of the CO concentrations" How about the mass balance?

Answer: Yes, correct. Same amounts of acetate were produced regardless of the CO concentration at high CO concentrations. This point was already raised by Rev. 1 and we have changed the sentence to: *in the absence of CO, the cells did not grow but they tolerated CO concentrations of 50, 75 and 100%*. This makes clear that grows is CO-dependent. Apparently, higher CO concentrations are only tolerated. We have not performed a mass balance during growth since the story here is that cells do grow on CO. A mass balance would be important for follow-up studies under controlled conditions in a fermenter. Mass balance of CO conversion in resting cells was measured and included in the revised manuscript.

15. Results, Line 138. "acetogenesis from CO also requires Na⁺/the Rnf complex". This seems important enough that the authors might wish to highlight in the abstract.

Answer: Good point, thank you! Since we added new data to the manuscript and since we have only 200 words for the abstract, we are afraid that there is no space left to include this statement.

16. Results, Line 169. “After ten transfers, the [Δ hydBA/hydA2] mutant grew on 100 mM formate with a similar growth rate as the Δ pyrE strain but produced twice the amount of biomass (final OD600 of 0.9) (Fig. 4a).” Did the level of hdcR increase? What is responsible for the higher OD?

Answer: We did not determine levels of HDCR since this would have no direct effect on cell yield. The cell yield is a direct reflection of the energetic status of the cells, and the energetic status is much improved due to the fact that electrons are no longer lost to the environment as molecular hydrogen, that formate oxidation only yields reduced ferredoxin, and that two thirds of the reduced ferredoxin are oxidized by the Rnf complex. As stated in the discussion, this increases the energetic status (ATP yield) by 300%.

17. Results. As far as I can tell, there is no information presented here that Fd is the redox mediator or which ferredoxin is involved in the hbcR reaction. If there is a hbcR isolated with the complex, has this been deleted? This is problematic throughout; for example, in the discussion, they make a number of apparent assumptions about Fd – i.e., Line 235 - ... indicating again that not CO toxicity but HDCR adaptation to ferredoxin”.

Answer: Again, a good point. Since this point is central to our conclusion it has been addressed experimentally and published for the HDCR from *A. woodii* as well as *T. kivui*. The purified enzymes, both can use ferredoxin as electron carrier *in vitro* (references are in the manuscript and also shown in this manuscript by the enzymatic assays in cell free extract). Whether ferredoxin can be used as electron carrier also *in vivo* can only be addressed by genetic studies. Such a mutant analyses was recently published by us for *T. kivui* (Dietrich & Müller, 2023, ACS Catalysis; cited in the manuscript) and in another paper published by us 5 weeks ago (Baum et al. Microbiol. Spectrum). The HDCR can use ferredoxin, but for an efficient, growth supportive effect, adaptations are required. As outlined in Baum et al. these adaptations occur in the electron wire subunits of the HDCR from *T. kivui*, most likely to increase the number of protomers of the HDCR which may be more accessible for ferredoxin than the long filaments and bundles. Our SNP analyses presented in the revised version come to the same conclusion. Both papers and the possible mechanism of adapting the HDCR to efficient use of ferredoxin are described in the revised version of the manuscript.

18. Results. Section on conversion of formate in resting cells of the CO-adapted Δ hydBA/hydA2 mutant. It seems problematic that the rates of formate conversion do not match that of acetate formation. The authors should explain this. How about the rate of CO₂ formation? CO formation?

Answer: We are a bit puzzled about this comment. If the reviewer refers to Figs 4 & 5, one can see that formate oxidation is slower in the Δ hydBA/hydA2 mutant, but acetate formation is also slowed down. The amount of formate oxidized matches very good the amount of acetate produced. In both strains, it was ca. 4 to 1, as expected.

19. Discussion. Line 214. The CO metabolism section should be more explicit. “Carbon monoxide metabolism in the Δ hydBA/hydA2 mutant is postulated to involve the HDCR.” Here they need a reference if it has been proposed below. Perhaps this is new. “In this case, reduced ferredoxin ...” Perhaps they mean CO dependent reduction of ferredoxin ...”

Answer: A reference for the statement is now given. We agree that “in this case” was not specific enough. The sentence is changed to “postulated to involve the HDCR (new reference Schwarz et al. 2020) and reduced ferredoxin is assumed to donate.....”

20. “In stark contrast, in the Δ hydBA/hydA2 mutant formate is oxidized with simultaneous reduction of ferredoxin (Fig. 6d).” However, Fig 6 doesn’t present results, but a scenario.

Answer: changed to “see Fig. 6d”.

21. Growth on CO seems to lead predominantly to formate, not acetate.

Answer: No, growing cells only produced acetate. Formate was not produced at all. Formate was only produced in resting cells, in which metabolism is uncoupled from growth. Under these conditions, the methyl branch of the Wood-Ljungdahl pathway apparently becomes the bottleneck and formate accumulates (often transiently). A transient accumulation of formate from $H_2 + CO_2$ has also been by others and was first described by the group of Ralf Conrad.

It seems possible that the major effect of the adaptation is relief from CO inhibition. Perhaps CO is converted to either CO_2 or formate, which are used for acetate (+?) formation.

Answer: Yes, exactly what we mean. CO is oxidized to CO_2 and reduced ferredoxin, and reduced ferredoxin then reduces CO_2 to formate in the HDCR enzyme. Formate is then further reduced to a methyl group and condensed with another CO, derived from reducing a mol of CO_2 , to acetate.

22. As shown in Suppl. Fig. 2, ethanol is also produced. it also seems that ethanol as well as acetate concentrations should have been determined during all the fermentations (especially on CO and formate) and it would be helpful to see a detailed mass/balance related to CO, CO_2 , acetate, ethanol and formate during the experiments. Also in a related issue, what happens during growth on CO + formate? Is formate preferentially utilized? Is the CO converted to formate which equilibrates with CO_2 as the substrate for acetogenesis.

Answer: Thank you for the comment which actually raises a number of different points.

Supplementary Fig. 2 shows grows on fructose, not CO or formate. Only with sugars, traces of ethanol are produced, and ethanol production from sugar by *A. woodii* has been characterized by us in detail and published before (Moon and Müller, 2021). During growth on CO, ethanol was NOT produced. Mass balances (substrate >product) have been presented in the manuscript for growth on formate which is exclusively converted to acetate.

Only resting cells, in which metabolism is uncoupled from growth, produce formate. If further conversion of formate, by inhibiting the formyl-THF synthetase reaction, is inhibited, formate is the only product. Since formate production is not ATP generating and NOT growth supportive, it only occurs in non-growing cells.

What happens during growth on CO + formate? Published biochemical analyses and mutant studies demonstrate, that CO is oxidized to CO_2 , and the electrons generated are used to reduce the formate added to a methyl group which is further converted to acetate with a carbonyl group directly derived from the added CO. Thus, formate is preferentially used as electron acceptor for CO oxidation. The CO_2 produced equilibrates with the CO_2 pool in the environment, but CO_2 is not required for acetogenesis from CO and formate according to:

Reviewer #3 (Remarks to the Author):

The manuscript of Moon and Mueller describes the development and characterization of a hydrogenase-free (with knockout of the two hydrogenases HydA2 and HydB) and CO-adapted mutant of the acetogenic bacterium *Acetobacterium woodii*. Unlike the wildtype, the mutant can grow on CO and formate and convert them to produce acetate and ATP. The hydrogenase-free $\Delta\text{hydBA/hydA2}$ mutant can directly use CO to generate formate and reduced ferredoxin as electron donor instead of H₂. The reduced ferredoxin can further fuel the respiratory enzymes ferredoxin:NAD-oxidoreductase (Rnf) complex that couples electron transfer to Na⁺-dependent ATP generation. Conversion of formate in resting cells of the CO-adapted $\Delta\text{hydBA/hydA2}$ mutant was quantitatively studied and compared to that of a ΔpyrE strain. This is an interesting work with significant advance regarding conversion of CO and formate by *A. woodii*.

Answer: Thank you for the comment.

However, there are some major issues that need to be addressed for further improving the quality of the work before publication.

1) In general, regarding the double deletion mutant which was adapted to grow on CO, the authors should perform genome re-sequencing to identify other possible mutations occurred during adaptation.

Answer: Thank you for the comment. As also requested by Reviewer 1, we added the genome sequencing results into the revised version of the manuscript in an extra, new chapter. The new data are also discussed in a new chapter in the revised version.

2) The roles of the other mutations should be clarified or at least discussed regarding their possible roles in affecting use of CO and formate and generation of energy (ATP).

Answer: Thank you for the comment. We described the roles of the other mutations with the genome sequencing results (see above).

3) L27, the ΔHydA2 variant is not described in the manuscript. The 25-fold formate production is achieved by $\Delta\text{HydBA/HydA2}$. More information about $\Delta\text{HydBA/HydA2}$ should be added.

Answer: Thank you for the comment. Here, ΔHydA2 variant means the ΔHydA2 variant of HDCR which is the $\Delta\text{hydBA/hydA2}$ mutant. This has been clarified in line 27 (see also comment of Rev. 1)

4) Line 50, “the calculated 1.5 ATP/acetate” which is shown in figure 6a, needs to be explained more clearly. And add reference(s).

Answer (same as to Rev. 1 for the same point): Both values come from the metabolic scenarios depicted in Fig. 6. The values used for the calculation (ions translocated /ATP; ions translocated per 2 electrons) have been experimentally determined or are calculated based on free energy changes of the redox reaction and the magnitude of the electrical potential across the cytoplasmic

membrane. The latter two values are referenced, the values for ATP/acetate are derived from Fig. 6.

5) L124, it is uncertain whether the additional bicarbonate will be decomposed into CO₂ and utilized as a carbon source? Why is the buffer different (bicarbonate or phosphate) when utilizing CO or formate?

Answer: Thank you for the comment. Yes, additional bicarbonate (60 mM) is in equilibrium with CO₂. Thus, it can be used as electron acceptor, in addition to the CO₂ produced from oxidation of the substrate CO. However, the effect that bicarbonate stimulates the production of formate is not due to formate production from CO₂, but to inhibition of subsequent formate reduction in the methyl branch of the WLP. This is due to an inhibition of the ATP synthase by bicarbonate (shown by us in Schwarz and Müller, 2020) which leads to a shortage of ATP and subsequently to an inhibition of ATP-dependent activation of formate to formyl-THF.

6) L133, why formate is also produced in the absence of bicarbonate? Pl give an explanation for this.

Answer: This is indeed not easy to explain, and we can only speculate. In the absence of CO₂/bicarbonate the CODH/ACS may be a bottleneck. Maybe the enzyme is more efficient in fusing CO₂-derived CO with a methyl group and coenzyme A to acetyl-CoA than using CO directly. Maybe CODH/ACS oxidises CO to CO₂ and produces a futile cycle in oxidizing CO and reducing CO₂? Since we really don't know the answer, we would like to refrain from an explanation in the text.

7) L527, why three quarters of formate are oxidized? Is it achieved from your calculation or reference? Is the oxidization ratio the same when using Fd₂- instead of H₂ as electron carrier as shown in figure.6d?

Answer: Thank you for the comment. Biochemistry and bioenergetics of formatotrophic acetogenesis in *A. woodii* of the wild type was already described in Moon *et al.*, 2021. In order to reduce one mol formate to acetate, 2 mol NADH and one mol reduced ferredoxin is required, in total 6 electrons. Reduced ferredoxin carries 2 electrons as H₂, therefore, 3 mol formate should be oxidized to CO₂ in the mutant as well.

8) L101, "a reductant other than H₂".

Answer: We changed this to an alternative reductant.

9) L134, about Fig.3c, it should be explained with more details.

Answer: see above, your point number 6.

10) L258, information about the Δ pyrE strain is missing and should be added.

Answer: a reference to the Δ pyrE is given in the revised version.

Reviewers' Comments:

Reviewer #1:

Remarks to the Author:

The authors diligently responded to all review comments we suggested, resulting in much clearer interpretation than the initial version. Thank you.

According to the Whole Genome Resequencing results of HDCR mutants, only *hycB2*, *modC2*, and *fdhC* seem to show significant changes. Among these, it appears difficult to determine how the mutation affected transporter function, especially for *modC2* and *fdhC*, which encode transporters.

The mutation in *hycB2*, as mentioned by the authors in the discussion section, seems to play a role in forming nanowires along with HDCR. However, with the deletion of HDCR, it does not seem that the two proteins forming the complex have a specific function. Therefore, interpreting the current phenotype based solely on WGS results seems challenging. Concluding that the deletion of HDCR enhances adaptation to CO would not seem problematic, so far.

While the findings of this study are highly interesting, there are still many aspects that remain unresolved. These areas require more data, and many are technically challenging at this time. With further research, we expect many of the questions raised in this paper to be addressed.

Reviewer #2:

Remarks to the Author:

The overall strength of the paper is that the authors reveal a seemingly straightforward and reproducible mechanism that *A. woodii* can recover from inhibition by CO. I feel that the authors addressed and satisfied most of my critiques. I felt relatively convinced by their argument that deletion of the H₂ases was key to the ability to grow on CO. The whole genome sequencing revealed several other potentially important mutations, including some in CODH/ACS. These are mentioned, but not highlighted in the manuscript and I did not see that the residues were identified. They should be as should all the other mutations. Their potential involvement in growth on CO should be addressed. I feel that the revisions do make the paper more accessible to the broader audience represented by Nature Communications. In addition, most of the ambiguities are clarified in the revision.

Table 1 legend: "CO oxidation yield Fd²⁻ only" change to "yields" – plural. Furthermore, it is highly unlikely that CO oxidation only couples to ferredoxin. It should be made clear that this is the direct coupling (and I doubt that that is true, since CODH is a very potent reductant that couples to various biological and nonbiological electron acceptors in vitro. It should be made clear that even if that statement were true, it is going to be the major electron donor in the cell for both anabolic and catabolic reactions. I'm not suggesting that they need to do any other experiments here, but the general audience should be informed about this. In addition, the CooS in the scheme is likely not the only CODH that reduces CO₂ to CO at the top of the cycle – clearly the CODH in the CODH/ACS complex can catalyze CO₂ reduction and is generally better at CO₂ reduction than most CooS enzymes.

Reviewer #3:

Remarks to the Author:

The authors have adequately addressed all my questions.

Point-by-point response to the reviewer's comments.

Reviewer #1 (Remarks to the Author):

The authors diligently responded to all review comments we suggested, resulting in much clearer interpretation than the initial version. Thank you.

Answer: Thank you for the comment.

According to the Whole Genome Resequencing results of HDCR mutants, only *hycB2*, *modC2*, and *fdhC* seem to show significant changes. Among these, it appears difficult to determine how the mutation affected transporter function, especially for *modC2* and *fdhC*, which encode transporters. The mutation in *hycB2*, as mentioned by the authors in the discussion section, seems to play a role in forming nanowires along with HDCR. However, with the deletion of HDCR, it does not seem that the two proteins forming the complex have a specific function. Therefore, interpreting the current phenotype based solely on WGS results seems challenging. Concluding that the deletion of HDCR enhances adaptation to CO would not seem problematic, so far.

While the findings of this study are highly interesting, there are still many aspects that remain unresolved. These areas require more data, and many are technically challenging at this time. With further research, we expect many of the questions raised in this paper to be addressed.

Answer: Thank you for your thoughts on this topic. We understand your points more as a general remark for future works, not meant to be addressed in this manuscript.

As observed very often, WGS does not provide a conclusive answer as to why the cells finally grew on CO. We agree that the role of ModC2 and FdhC in adaptation is not obvious and, most important, they were not found in 100% of the samples. However, the *hycB2* mutation occurred in 100% of the samples and the mutation is also plausible, as discussed in the manuscript. We should point out that not the entire HDCR (as stated by the reviewer) was deleted, but only HydA2! FdhF, HycB2 and HycB3 are still present! So, our discussion that an altered HycB2/HycB3 interaction favours access to ferredoxin is valid.

Anyway, we do not hear critique that we should address. Of course, as mentioned, follow up stories on this interesting observation will be performed in the future.

Reviewer #2 (Remarks to the Author):

The overall strength of the paper is that the authors reveal a seemingly straightforward and reproducible mechanism that *A. woodii* can recover from inhibition by CO. I feel that the authors addressed and satisfied most of my critiques. I felt relatively convinced by their argument that deletion of the H2ases was key to the ability to grow on CO.

Answer: Thank you for the comment.

The whole genome sequencing revealed several other potentially important mutations, including some in CODH/ACS. These are mentioned, but not highlighted in the manuscript and I did not see that the

residues were identified. They should be as should all the other mutations. Their potential involvement in growth on CO should be addressed.

Answer: There seems to be a misunderstanding. The residues have been identified and they could/and still can be found in Table 1. We found one mutation in only 5.3% of AcsA, and two others with only 8.7% and 21.8% in AcsB. Since this is a relatively small portion, we do not consider these mutations as being important for the observed phenotype and, therefore, did not describe them in further detail in the manuscript.

I feel that the revisions do make the paper more accessible to the broader audience represented by Nature Communications. In addition, most of the ambiguities are clarified in the revision.

Answer: Thank you for the comment.

Table 1 legend: “CO oxidation yield Fd²⁻ only” change to “yields” – plural.

Answer: Corrected.

Furthermore, it is highly unlikely that CO oxidation only couples to ferredoxin. It should be made clear that this is the direct coupling (and I doubt that that is true, since CODH is a very potent reductant that couples to various biological and nonbiological electron acceptors in vitro).

Answer: We disagree with this notion. There is overwhelming evidence from different groups on CO dehydrogenases from different sources that ferredoxin is the physiological electron donor, not NAD or NADP or flavins. Thus, this is considered textbook knowledge.

It should be made clear that even if that statement were true, it is going to be the major electron donor in the cell for both anabolic and catabolic reactions. I’m not suggesting that they need to do any other experiments here, but the general audience should be informed about this.

Answer: We have included the sentence: “and Fd²⁻ is a major electron donor for anabolic and catabolic reactions” into the legend of Fig. 1.

In addition, the CooS in the scheme is likely not the only CODH that reduces CO₂ to CO at the top of the cycle – clearly the CODH in the CODH/ACS complex can catalyze CO₂ reduction and is generally better at CO₂ reduction than most CooS enzymes.

Answer: There seems to be a misunderstanding. The CO-oxidizing enzyme is indicated in Fig. 1 as CooS, the CO₂-reducing enzyme as CODH/ACS. Although both enzymes are reversible *in vitro*, the cellular function (as delineated by genetic studies) of CooS is oxidation of the substrate CO whereas the function of CODH/ACS is reduction of CO₂ to CO in the carbonyl branch of the WLP.

Reviewer #3 (Remarks to the Author):

The authors have adequately addressed all my questions.

Answer: Thank you for the comment.

Reviewers' Comments:

Reviewer #1:

Remarks to the Author:

The authors have adequately addressed all my questions.

Reviewer #2:

Remarks to the Author:

My major concern is the dogmatic statement "... ferredoxin is the physiological electron donor, not NAD or NADP". Agreeably, it has been shown that CODH does not directly transfer electrons to NAD or NADP or vice versa for NAD(P)H. Flavodoxin clearly substitutes for ferredoxin under low iron conditions in systems where Ferredoxin is expressed at very low levels. I just feel the authors are propagating statements that are at odds with some rigorous studies. But I also don't want to belabor this point, which doesn't make or break acceptance of this otherwise good paper.

Point-by-point response to the reviewer's comments.

Reviewer #1 (Remarks to the Author):

The authors have adequately addressed all my questions.

Answer: Thank you for the comment.

Reviewer #2 (Remarks to the Author):

My major concern is the dogmatic statement "... ferredoxin is the physiological electron donor, not NAD or NADP". Agreeably, it has been shown that CODH does not directly transfer electrons to NAD or NADP or vice versa for NAD(P)H. Flavodoxin clearly substitutes for ferredoxin under low iron conditions in systems where Ferredoxin is expressed at very low levels. I just feel the authors are propagating statements that are at odds with some rigorous studies. But I also don't want to belabor this point, which doesn't make or break acceptance of this otherwise good paper.

Answer: As mentioned before, we do not see a problem with this statement for it is textbook knowledge. Thanks for your understanding!